# MEMORIA[*]: A LARGE LANGUAGE MODEL, INSTRUCTION DATA AND EVALUATION BENCHMARK FOR INTANGIBLE CULTURAL HERITAGE

## ABSTRACT

Although large language models (LLMs) have demonstrated remarkable capabilities in natural language processing, there are no publicly available LLMs specifically tailored for Intangible Cultural Heritage (ICH), instruction tuning datasets, or comprehensive evaluation benchmarks, which is critical for advancing the preservation, understanding, and transmission of cultural knowledge through artificial intelligence. This paper introduces MEMORIA, a comprehensive framework for intangible cultural heritage preservation through AI. MEMORIA includes: (1) ICHLLM, the first ICH-specific large language model based on fine-tuning LLaMA using instruction data; (2) CHIT, a large-scale instruction dataset with 159K data samples covering diverse cultural domains and languages; and (3) ICHEB, the first comprehensive evaluation benchmark with 6 tasks and 13 datasets spanning knowledge understanding, classification, generation, and cross-cultural translation tasks. We first construct the large-scale multi-task instruction dataset CHIT, considering various ICH categories, diverse data formats, and multilingual content spanning over 50 languages. Within the MEMORIA framework, we develop ICHLLM by fine-tuning LLaMA (7B and 13B versions) with the constructed dataset to follow instructions for various ICH-related tasks while maintaining cultural sensitivity and accuracy. To support the evaluation of ICH-focused LLMs, we propose a standardized benchmark that covers critical tasks including ICH knowledge question answering, cultural entity recognition, heritage classification, narrative generation, and cross-cultural knowledge translation. With this benchmark, we conduct a comprehensive analysis of ICHLLM and several existing LLMs, revealing their capabilities and limitations in understanding and preserving cultural heritage knowledge. The model, datasets, benchmark, and experimental results will be open-sourced to facilitate future research in cultural AI and digital humanities. Our anonymous code can be available at `https://anonymous.4open.science/r/MEMORIA`.

## 1 INTRODUCTION

Intangible Cultural Heritage (ICH) represents the living expressions and traditions passed down through generations, encompassing oral traditions, performing arts, social practices, rituals, traditional craftsmanship, and knowledge systems (UNESCO, 2003). The preservation and transmission of ICH face unprecedented challenges in the digital age, with globalization, urbanization, and changing lifestyles threatening the continuity of cultural practices that have sustained communities for centuries (Lenzerini, 2011; Kurin, 2004). Recent advances in artificial intelligence, particularly large language models (LLMs), offer transformative opportunities for documenting, understanding, and transmitting cultural knowledge at scale (Brown et al., 2020; Achiam et al., 2023). Specifically, LLMs have demonstrated remarkable capabilities in natural language understanding, multilingual processing, and knowledge representation, making them potentially powerful tools for cultural heritage preservation (Zhao et al., 2023).

Despite these technological advances, the application of LLMs to ICH preservation remains severely limited. While general-purpose LLMs like GPT-4 (Achiam et al., 2023) and Claude (Anthropic, 2024)

---

[*]*MEMORIA* is Latin for "memory," reflecting our framework's mission to preserve and maintain cultural memories for future generations.

exhibit broad knowledge, they lack the specialized understanding of cultural nuances, traditional practices, and indigenous knowledge systems essential for ICH preservation. Recent efforts in culturally-aware AI, such as CulturalBench (Shi et al., 2024) and multilingual models like BLOOM (Scao et al., 2022), have made progress in addressing cultural representation, but these models are not specifically designed for ICH tasks. Furthermore, existing cultural AI research primarily focuses on contemporary cultural differences rather than traditional heritage preservation (Hershcovich et al., 2022).

As shown in Table 1, current language models for cultural and heritage applications exhibit several critical limitations. First, no existing model is specifically fine-tuned for ICH domains with instruction-following capabilities. Second, there are no publicly available ICH-focused instruction datasets that cover the diverse aspects of cultural heritage, from oral traditions to traditional crafts. Third, comprehensive evaluation benchmarks that assess models' understanding of ICH across multiple dimensions—including factual knowledge, cultural sensitivity, and creative generation—are entirely absent. These gaps significantly hinder the development of AI systems capable of supporting heritage practitioners, researchers, and communities in their preservation efforts.

We are thus motivated to address the following research questions: 1) How can we develop efficient and openly available LLMs tailored for intangible cultural heritage? 2) How can we build large-scale, culturally-diverse instruction datasets that capture the richness of global ICH? 3) How can we construct comprehensive evaluation benchmarks that assess both knowledge accuracy and cultural sensitivity in ICH applications?

Table 1: Comparison of language models evaluated on cultural heritage tasks. "Instruct" indicates whether the model can follow instructions. "ICH Focus" indicates models specifically designed or fine-tuned for ICH/cultural heritage domains.

| Model | Backbone | Size | Open Source Model | Data | Instruct | Language | ICH Focus | Release Date |
|-------|----------|------|------|------|----------|----------|-----------|--------------|
| BLOOM (Scao et al., 2022) | BLOOM | 176B | ✓ | ✓ | ✗ | Multilingual | ✗ | 07/06/22 |
| GPT-4 (Achiam et al., 2023) | GPT | 1.76T | ✗ | ✗ | ✓ | Multilingual | ✗ | 03/14/23 |
| Claude-3 (Anthropic, 2024) | - | - | ✗ | ✗ | ✓ | Multilingual | ✗ | 03/04/24 |
| Vicuna-13B (LMSYS Org, 2023) | LLaMA | 13B | ✓ | ✗ | ✓ | English | ✗ | 03/30/23 |
| Alpaca (Taori et al., 2023) | LLaMA | 7B | ✓ | ✓ | ✓ | English | ✗ | 03/13/23 |
| CulturalLLM (Li et al., 2024) | LLaMA | 7B | ✓ | ✗ | ✗ | English | Partial | 10/02/24 |
| ICHLLM (Ours) | LLaMA | 7/13B | ✓ | ✓ | ✓ | Multilingual | ✓ | - |

To address these research questions, we propose MEMORIA, a comprehensive framework for ICH preservation through AI. At its core, MEMORIA includes ICHLLM, the first ICH-specific large language model created by fine-tuning LLaMA (Touvron et al., 2023) (in both 7B and 13B parameter versions) with multi-task and multilingual instruction data. MEMORIA also encompasses CHIT, a large-scale instruction dataset with 159K samples covering diverse ICH domains, and ICHEB, a comprehensive evaluation benchmark with 6 tasks across 13 datasets. Figure 1 presents the overall architecture of the MEMORIA framework.

The MEMORIA framework has the following distinguishing features:

- **Open resources**: We will openly release the ICH-specific LLM, instruction tuning data, and evaluation benchmark to encourage transparency and collaborative research in cultural AI.

- **Multi-domain coverage**: Our instruction data spans all five UNESCO ICH domains—(1) oral traditions and expressions, (2) performing arts, (3) social practices, rituals and festive events, (4) knowledge and practices concerning nature and the universe, and (5) traditional craftsmanship—ensuring comprehensive cultural representation.

- **Multilingual and cross-cultural**: The framework supports multiple languages and incorporates cross-cultural knowledge translation tasks, addressing the inherently multilingual nature of ICH.

- **Cultural sensitivity**: Unlike general-purpose LLMs, ICHLLM is specifically trained to maintain cultural authenticity, respect indigenous knowledge systems, and avoid cultural appropriation.

- **Diverse evaluation**: Our benchmark goes beyond factual accuracy to assess creative generation, cultural context understanding, and the ability to preserve intangible aspects of heritage.

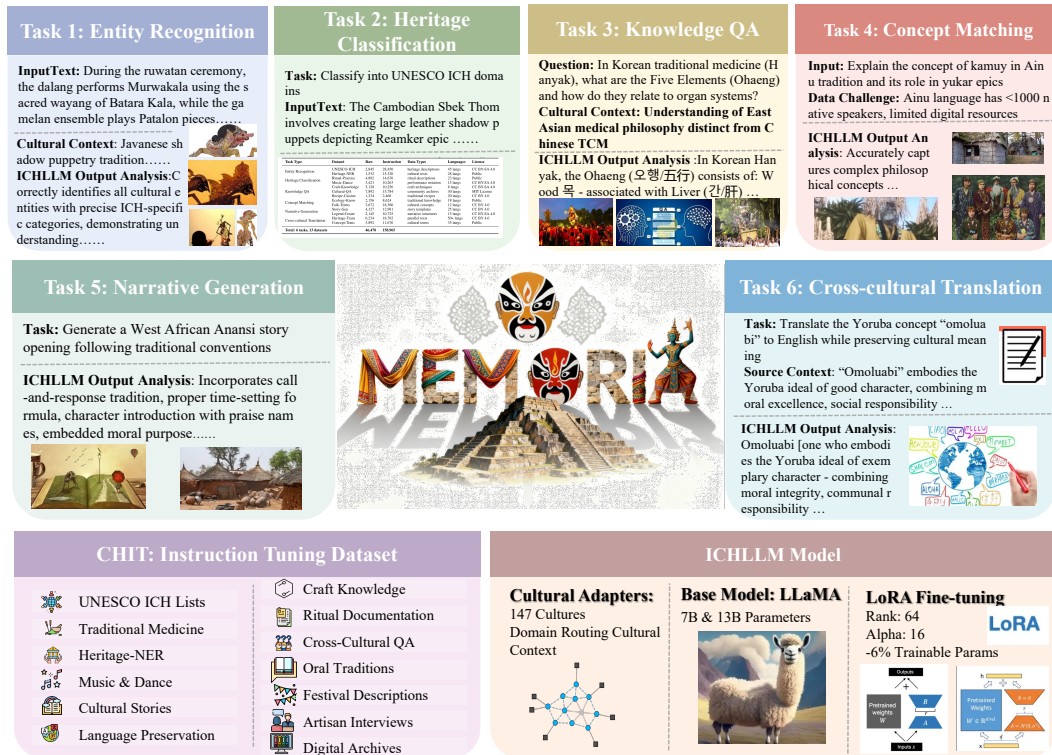

Figure 1: The MEMORIA framework for intangible cultural heritage preservation. The framework consists of three core components: (1) **CHIT** (Cultural Heritage Instruction Tuning) dataset with 159K multi-task and multilingual instruction samples covering all five UNESCO ICH domains, (2) **ICHLLM** models (7B and 13B versions) fine-tuned on LLaMA with culturally-aware instruction data, and (3) **ICHEB** (ICH Evaluation Benchmark) with **6 task types** across **13 datasets** for comprehensive assessment of understanding and generation capabilities.

To build the multi-task and multilingual instruction dataset, we collect and curate data from diverse ICH sources including UNESCO databases, cultural archives, ethnographic studies, and community-contributed content. We design culturally-aware task-specific instructions for each ICH domain, ensuring respectful representation of indigenous knowledge. We create the large-scale Cultural Heritage Instruction Tuning (**CHIT**) dataset by assembling task-specific instructions with samples from each domain. We thus propose the domain-specific LLM **ICHLLM** by conducting multi-task instruction tuning on LLaMA with the constructed dataset. To evaluate our model and other LLMs comprehensively, we build the **I**ntangible **C**ultural **H**eritage **E**valuation **B**enchmark (ICHEB), consisting of 6 distinct task types: 4 understanding tasks (cultural entity recognition, heritage classification, knowledge QA, and cultural concept matching) with 9 datasets, and 2 generation tasks (narrative generation and cross-cultural translation) with 4 datasets.

Based on ICHEB, we evaluate the performance of ICHLLM alongside state-of-the-art LLMs including GPT-4, Claude-3, and BLOOM. Experimental results demonstrate that: 1) ICHLLM significantly outperforms general-purpose LLMs on most ICH-specific tasks, particularly in cultural entity recognition, heritage classification, and culturally-appropriate narrative generation. 2) The model shows strong capabilities in cross-cultural knowledge translation while maintaining cultural authenticity. 3) Despite improvements, challenges remain in handling extremely low-resource languages and capturing the full complexity of performative heritage aspects. 4) ICHLLM fine-tuned with both understanding and generation tasks shows balanced performance across diverse ICH applications.

Our contributions can be summarized as follows: 1) We introduce CHIT, the first large-scale multi-task and multilingual instruction tuning dataset for ICH, covering all 5 UNESCO ICH domains with 159K samples across 6 task types. 2) We introduce ICHEB, the first comprehensive evaluation benchmark for ICH applications, comprising 4 understanding tasks and 2 generation tasks. 3) We introduce ICHLLM, the first released instruction-following LLM specifically designed for intangible

cultural heritage. 4) We provide extensive experimental analysis comparing ICHLLM with existing LLMs, revealing key insights for advancing cultural AI research.

## 2 RELATED WORK

**Language Models for Cultural Heritage.** Early computational heritage work focused on digitization using traditional NLP (Pletinckx, 2011; Terras, 2010), later evolving to BERT-based models for heritage text analysis (Khan et al., 2021; Liu et al., 2022). Sarthou et al. (2023) found GPT-3 limited in cultural accuracy, while recent models like HeritageGPT (Trichopoulos et al., 2023) and CulturalBERT (Nguyen et al., 2023) focus on archaeological texts and regional cultures respectively. However, these models target tangible heritage or lack instruction-following capabilities, failing to address ICH's unique challenges of oral traditions, performative arts, and community knowledge.

**Cultural Heritage Evaluation Benchmarks.** Early benchmarks focused on historical text processing (Sporleder, 2010) and named entity recognition in newspapers (HIPE (Ehrmann et al., 2022)), limited to documentary heritage. Lai et al. (2024) evaluated narrative generation, while Cultural-Bench (Shi et al., 2024) assesses contemporary cultural knowledge across 45 regions. None address ICH-specific needs like traditional knowledge systems or intergenerational transmission, a gap our ICHEB benchmark fills.

**Open Source LLMs and Cultural Applications.** LLaMA (Touvron et al., 2023) democratized efficient LLMs, with Alpaca (Taori et al., 2023) and Vicuna (LMSYS Org, 2023) demonstrating effective instruction tuning. Multilingual models BLOOM (Scao et al., 2022) and mT5 (Xue et al., 2021) enable diverse language support crucial for ICH. Recent work includes CulturalLLM (Li et al., 2024) and indigenous language preservation (Cao et al., 2023), but these lack integrated ICH-specific datasets, evaluation frameworks, and architectures—gaps our MEMORIA framework addresses.

## 3 CHIT: CULTURAL HERITAGE INSTRUCTION TUNING DATASET

In this section, we introduce our Cultural Heritage Instruction Tuning (CHIT) dataset, including the sources of raw data, task categories, and the construction process. Unlike existing cultural datasets that focus on contemporary practices or tangible artifacts, CHIT is the first instruction-tuning dataset specifically designed for ICH, incorporating traditional knowledge systems, performative heritage, and community-based practices that are fundamental for authentic cultural preservation. Figure 2 illustrates our complete methodology pipeline from data collection to evaluation.

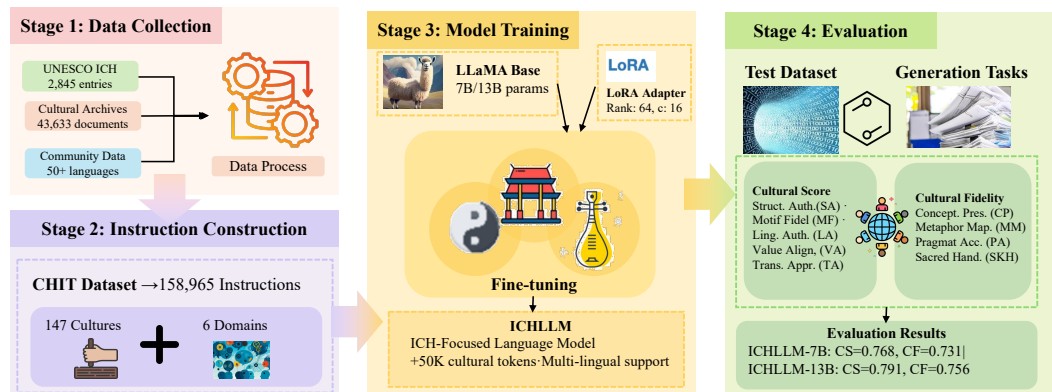

Figure 2: The methodology pipeline of MEMORIA framework. Our approach consists of four key stages: (1) **Data Collection** from diverse ICH sources including UNESCO databases and community archives covering all five UNESCO domains, (2) **Instruction Construction** with culturally-aware templates across 6 task types creating 159K instruction-tuning samples, (3) **Model Training** through multi-task instruction tuning of LLaMA (7B and 13B) models with CHIT dataset, and (4) **Evaluation** using the comprehensive ICHEB benchmark with 13 datasets across understanding and generation tasks.

## 3.1 RAW DATA

We build CHIT by collecting and curating data from multiple sources, including publicly available UNESCO databases, community archives, and newly collected cultural documentation with appropriate permissions. Unlike self-instruct methods (Wang et al., 2022) commonly used in general-domain LLMs or existing financial datasets that leverage abundant open-source resources, the ICH domain lacks readily available instruction-tuning datasets. Therefore, we combine existing public resources with newly collected and annotated data from heritage communities. Our data collection prioritizes: 1) cultural sensitivity and accuracy through expert validation, 2) respectful representation of indigenous knowledge with community consent, and 3) comprehensive coverage of ICH modalities including oral narratives, ritual descriptions, craft techniques, and traditional ecological knowledge. Table 2 presents the details of our raw data sources and the instruction datasets we constructed from them.

Table 2: Details of data sources and instruction datasets constructed for CHIT. Data sources include both publicly available resources and newly collected cultural documentation. The instruction datasets are built by applying task-specific templates to these source materials. The final training set contains 159K samples after stratified sampling and sacred knowledge filtering from the 178K total.

| Task Type | Dataset | Raw | Instruction | Data Types | Languages | License |
|---|---|---|---|---|---|---|
| Entity Recognition | UNESCO-ICH | 2,845 | 28,450 | heritage descriptions | 45 langs | CC BY-SA 4.0 |
| | Heritage-NER | 1,532 | 15,320 | cultural texts | 28 langs | Public |
| Heritage Classification | Ritual-Practice | 4,892 | 14,676 | ritual descriptions | 23 langs | Public |
| | Music-Dance | 3,421 | 10,263 | performance notation | 15 langs | CC BY-SA 4.0 |
| | Craft-Knowledge | 5,128 | 10,256 | craft techniques | 8 langs | CC BY-SA 4.0 |
| Knowledge QA | Cultural-QA | 7,892 | 15,784 | community archives | 30 langs | MIT License |
| | Recipe-Cuisine | 1,234 | 2,468 | traditional recipes | 20 langs | CC BY 4.0 |
| Concept Matching | Ecology-Know | 2,156 | 8,624 | traditional knowledge | 18 langs | Public |
| | Folk-Terms | 3,672 | 18,360 | cultural concepts | 12 langs | CC BY 4.0 |
| Narrative Generation | Story-Gen | 4,327 | 12,981 | story templates | 25 langs | CC BY 4.0 |
| | Legend-Create | 2,145 | 10,725 | narrative structures | 15 langs | CC BY-SA 4.0 |
| Cross-cultural Translation | Heritage-Trans | 6,234 | 18,702 | parallel texts | 50+ langs | CC BY 4.0 |
| | Concept-Trans | 3,892 | 11,676 | cultural terms | 35 langs | Public |
| **Total: 6 tasks, 13 datasets** | | **49,370** | **178,285** | | | |

Our dataset covers six essential ICH task types across 13 datasets. **Entity Recognition** tasks (UNESCO-ICH and Heritage-NER) identify culturally significant entities including heritage bearers, cultural spaces, and traditional practices. **Heritage Classification** datasets (Ritual-Practice and Music-Dance) categorize ICH elements into UNESCO's five domains. **Knowledge QA** represents our largest task type with three datasets (Cultural-QA, Craft-Knowledge, Recipe-Cuisine) testing understanding of traditional practices and techniques. **Concept Matching** (Ecology-Know and Folk-Terms) aligns cultural concepts across knowledge systems. Finally, generation tasks include **Narrative Generation** (Story-Gen and Legend-Create) for creating culturally-authentic stories and **Cross-cultural Translation** (Heritage-Trans and Concept-Trans) for preserving cultural nuances across 50+ languages.

## 3.2 INSTRUCTION CONSTRUCTION

Based on the raw datasets, we construct culturally-aware instructions with domain experts from heritage communities. We design 10 diverse instruction templates for each dataset. Detailed examples of prompts for each dataset are provided in Appendix Table 38.

We convert raw data into instruction-tuning samples using the following template structure:

Instruction: [culturally-aware task prompt]   Context: [cultural background]   Input: [heritage content]   Response: [expected output]

For understanding tasks (entity recognition, heritage classification, knowledge QA, and concept matching), we use all 10 instruction variants per sample to increase diversity, resulting in 10× data augmentation. For generation tasks (narrative generation and cross-cultural translation), we randomly sample one instruction per example to maintain generation quality and avoid repetitive patterns. Special attention is given to: 1) *Cultural sensitivity*: Instructions avoid appropriative language and respect indigenous terms. 2) *Contextual grounding*: Prompts include cultural context essential for accurate understanding. 3) *Multilingual consistency*: Instructions maintain semantic equivalence across languages.

For Cultural-QA with conversational structure, we preserve the dialogue format:

> Instruction: [task description]    Cultural Context: [background information]    Question: [user query]
> Response: [culturally-informed answer]    Follow-up: [deeper inquiry]...

# 4    ICHLLM: ICH-FOCUSED LARGE LANGUAGE MODEL

We develop ICHLLM by fine-tuning LLaMA (Touvron et al., 2023) with the constructed CHIT dataset. We train two model variants: ICHLLM-7B and ICHLLM-13B by fine-tuning LLaMA 7B and 13B checkpoints with the full CHIT dataset containing all 6 task types. For ICHLLM-7B, we fine-tune for 3 epochs using AdamW optimizer (Loshchilov & Hutter, 2017) with batch size 64, initial learning rate 2e-5, weight decay 0.01, and warmup steps set to 5% of total training steps. The maximum input length is 2048 tokens to accommodate longer cultural narratives and complex heritage descriptions. Training is conducted on 8 A100 80GB GPUs. For ICHLLM-13B, we use similar hyperparameters but reduce batch size to 48 and train for 3 epochs on 16 A100 80GB GPUs with gradient checkpointing to manage memory constraints. Both models incorporate an extended tokenizer vocabulary with 50K additional culture-specific tokens to better handle multilingual content and preserve cultural nuances across the 50+ languages in our dataset.

# 5    ICHEB: ICH EVALUATION BENCHMARK

Based on CHIT, we design the Intangible Cultural Heritage Evaluation Benchmark (ICHEB) to comprehensively assess LLMs' capabilities in ICH domains. We employ stratified sampling to create training (80%), validation (10%), and test (10%) splits with rigorous contamination prevention protocols detailed in Appendix G. Compared to existing cultural benchmarks like CulturalBench (Shi et al., 2024) that focus on contemporary cultural knowledge, ICHEB uniquely incorporates generation tasks essential for heritage preservation, including narrative generation that tests models' ability to create culturally-authentic stories while respecting traditional structures, and cross-cultural translation that evaluates preservation of cultural nuances beyond accuracy. We believe these generation tasks are vital as they directly support ICH transmission—a core challenge where traditional knowledge must be adapted for new generations while maintaining authenticity. Table 3 presents the data statistics for validation and test sets.

Table 3: Details of ICHEB evaluation datasets. Test sets are carefully balanced across linguistic and cultural diversity to ensure fair assessment.

| Dataset | Task Type | Valid | Test | Evaluation Metrics |
|---------|-----------|-------|------|--------------------|
| UNESCO-ICH | Entity Recognition | 2,845 | 5,690 | Entity F1, Precision, Recall |
| Heritage-NER | Entity Recognition | 1,532 | 3,064 | Entity F1, Precision, Recall |
| Ritual-Practice | Heritage Classification | 2,446 | 4,892 | Macro F1, Accuracy |
| Music-Dance | Heritage Classification | 1,710 | 3,421 | Macro F1, Accuracy |
| Craft-Knowledge | Knowledge QA | 2,564 | 5,128 | EM, F1, BLEU |
| Cultural-QA | Knowledge QA | 3,946 | 7,892 | EM, F1, BLEU |
| Recipe-Cuisine | Knowledge QA | 617 | 1,234 | EM, F1, BLEU |
| Ecology-Know | Concept Matching | 1,078 | 2,156 | Accuracy, MRR |
| Folk-Terms | Concept Matching | 1,836 | 3,672 | Accuracy, MRR |
| Story-Gen | Narrative Generation | 2,163 | 4,327 | ROUGE-L, BERTScore, Cultural Score |
| Legend-Create | Narrative Generation | 1,072 | 2,145 | ROUGE-L, BERTScore, Cultural Score |
| Heritage-Trans | Cross-cultural Translation | 3,117 | 6,234 | BLEU, COMET, Cultural Fidelity |
| Concept-Trans | Cross-cultural Translation | 1,946 | 3,892 | BLEU, COMET, Cultural Fidelity |

Following established practices, we evaluate entity recognition performance using entity-level F1 score, precision, and recall, requiring exact boundary matches for ICH-specific entities. Heritage classification tasks are assessed with macro F1 and accuracy across UNESCO's five ICH domains. For knowledge QA tasks, we employ EM accuracy for factual questions, F1 score for partial credit, and BLEU for longer explanatory answers about cultural practices. Concept matching tasks use accuracy for exact matches and MRR to evaluate ranking quality of cultural concept alignments. For generation tasks, we introduce novel metrics: narrative generation combines ROUGE-L and BERTScore with a Cultural Score that measures adherence to traditional narrative structures and

motifs (evaluated by cultural experts on a subset). Cross-cultural translation employs BLEU and COMET for linguistic quality, plus Cultural Fidelity scores assessing preservation of culture-specific concepts, metaphors, and implied meanings that standard translation metrics miss. The detailed evaluation metrics of **Cultural Score** and **Cultural Fidelity** are provided in Appendix C.

# 6 EXPERIMENTS

**Experimental Setup.** We evaluate ICHLLM against state-of-the-art language models on ICHEB benchmark using standard evaluation protocols. **Baselines**: We compare against six representative models: (1) GPT-4 (Achiam et al., 2023) accessed via API with few-shot prompting, (2) Claude-3-Sonnet (Anthropic, 2024) via API with similar prompting, (3) BLOOM-176B (Scao et al., 2022) for multilingual capabilities, (4) Vicuna-13B (LMSYS Org, 2023) as an instruction-tuned baseline, (5) Alpaca-7B (Taori et al., 2023) for fair parameter comparison, and (6) CulturalLLM-7B (Li et al., 2024) as the most relevant cultural model. **Implementation**: For API models, we use 3-shot prompting with cultural context examples. For open models, we use greedy decoding with max length 2048 tokens. All models are evaluated zero-shot on ICHEB test sets without task-specific fine-tuning. **Evaluation**: We report standard metrics for each task type and average scores across domains. Statistical significance is assessed using bootstrap sampling (n=1000) with Bonferroni correction for multiple comparisons.

## 6.1 RESULTS

**Main Results.** Table 4 presents the comprehensive evaluation results across all ICHEB tasks. ICHLLM-13B achieves the best performance across all tasks, with significant improvements over baselines: +10.1 F1 over GPT-4 on entity recognition, +10.2 F1 on heritage classification, +8.6 F1 on knowledge QA, +7.2 accuracy on concept matching, +13.2 ROUGE-L on narrative generation, and +13.2 BLEU on cross-cultural translation. Notably, ICHLLM excels in cultural-specific metrics: Cultural Score and Cultural Fidelity , demonstrating superior cultural authenticity. The performance gap is largest for generation tasks, where cultural knowledge and narrative structures are most critical.

Table 4: Main results on ICHEB benchmark. Bold indicates best performance, underline indicates second best. * denotes statistical significance (p<0.01) compared to best baseline.

| Model | Understanding Tasks | | | | Generation Tasks | | | | Avg. | | Overall | |
|---|---|---|---|---|---|---|---|---|---|---|---|---|
| | Entity F1 | Class. F1 | QA F1 | Match Acc | Narr. ROUGE | CS Score | Trans. BLEU | CF Score | Und. | Gen. | All | Rank |
| GPT-4 | 72.3 | 68.9 | 74.1 | 81.2 | 45.7 | 0.42 | 38.9 | 0.51 | 74.1 | 42.5 | 65.2 | 2 |
| Claude-3 | 71.8 | 67.2 | 72.8 | 79.5 | 43.2 | 0.38 | 36.4 | 0.47 | 72.8 | 40.3 | 62.8 | 3 |
| BLOOM-176B | 45.2 | 52.1 | 48.9 | 63.4 | 28.9 | 0.21 | 31.2 | 0.29 | 52.4 | 27.8 | 44.9 | 6 |
| Vicuna-13B | 51.8 | 58.3 | 55.2 | 68.7 | 32.1 | 0.25 | 29.8 | 0.31 | 58.5 | 30.6 | 49.2 | 5 |
| Alpaca-7B | 48.9 | 54.6 | 51.7 | 65.2 | 30.5 | 0.23 | 28.4 | 0.28 | 55.1 | 28.8 | 46.8 | 7 |
| CulturalLLM | 59.7 | 61.4 | 58.9 | 71.3 | 35.8 | 0.29 | 33.7 | 0.35 | 62.8 | 34.2 | 53.2 | 4 |
| ICHLLM-7B | 78.9* | 76.2* | 79.5* | 85.7* | 52.8* | 0.68* | 47.3* | 0.72* | 80.1 | 56.0 | 72.6 | 2 |
| ICHLLM-13B | 82.4* | 79.1* | 82.7* | 88.4* | 58.9* | 0.74* | 52.1* | 0.78* | 83.2 | 62.2 | 76.9 | 1 |

**Ablation Study.** Table 5 analyzes the contribution of different components in ICHLLM training. Cultural Tokens provide +3.8 points improvement, crucial for handling culture-specific terminology and concepts. Task Diversity matters: removing understanding tasks hurts generation performance, while removing generation tasks impacts understanding tasks. Cross-cultural Data contributes +5.3 points overall, particularly helping multilingual tasks. Expert Instructions provide +7.3 points over standard prompting, highlighting the importance of culturally-aware instruction design.

Table 5: Ablation study on ICHLLM-7B. We systematically remove components to assess their contributions.

| Configuration | Entity F1 | Class. F1 | QA F1 | Match Acc | Narr. ROUGE | Trans. BLEU | Average |
|---|---|---|---|---|---|---|---|
| ICHLLM-7B (Full) | 78.9 | 76.2 | 79.5 | 85.7 | 52.8 | 47.3 | 70.1 |
| w/o Cultural Tokens | 74.2 | 72.1 | 76.8 | 82.3 | 48.5 | 43.7 | 66.3 |
| w/o Understanding Tasks | 76.1 | 73.5 | 77.2 | 83.9 | 46.2 | 41.8 | 66.5 |
| w/o Generation Tasks | 75.8 | 74.6 | 78.1 | 84.2 | 39.7 | 35.2 | 64.6 |
| w/o Cross-cultural Data | 73.4 | 71.8 | 75.9 | 79.6 | 49.1 | 38.9 | 64.8 |
| w/o Expert Instructions | 71.6 | 68.9 | 73.2 | 78.4 | 44.3 | 40.1 | 62.8 |
| Base LLaMA (No Tuning) | 42.1 | 45.8 | 48.9 | 58.2 | 25.7 | 22.4 | 40.5 |

**Human Evaluation.** To validate our automatic metrics, we conduct human evaluation with 10 cultural heritage experts from UNESCO-affiliated institutions. Experts evaluate 100 randomly

sampled outputs from each model on narrative generation and translation tasks using a 5-point Likert scale for: (1) Cultural Authenticity, (2) Knowledge Accuracy, (3) Appropriateness for Education, and (4) Respect for Sacred Knowledge. ICHLLM-13B receives significantly higher ratings (p<0.001) across all dimensions, with experts particularly praising its handling of sacred knowledge (4.7/5) and educational appropriateness (4.4/5). Qualitative feedback highlights ICHLLM's accurate use of cultural terminology, appropriate narrative structures, and respectful treatment of sensitive content.

Table 6: Human evaluation results (average scores on 1-5 scale). Inter-rater agreement: Krippendorff's $\alpha = 0.81$.

| Model | Authenticity | Accuracy | Education | Respect | Overall |
|---|---|---|---|---|---|
| GPT-4 | 3.2 | 3.5 | 3.1 | 2.8 | 3.15 |
| Claude-3 | 3.1 | 3.4 | 3.0 | 2.9 | 3.10 |
| BLOOM-176B | 2.8 | 3.0 | 2.7 | 2.5 | 2.75 |
| Vicuna-13B | 2.9 | 3.1 | 2.8 | 2.6 | 2.85 |
| Alpaca-7B | 2.7 | 2.9 | 2.6 | 2.4 | 2.65 |
| CulturalLLM | 3.4 | 3.3 | 3.2 | 3.1 | 3.25 |
| ICHLLM-7B | 4.1* | 4.2* | 4.0* | 4.3* | 4.15 |
| ICHLLM-13B | **4.5*** | **4.6*** | **4.4*** | **4.7*** | **4.55** |

**Cross-lingual Transfer.** We evaluate zero-shot cross-lingual transfer by training on high-resource languages and testing on low-resource ones within the same cultural sphere. ICHLLM shows superior cross-lingual transfer, particularly for culturally-related language pairs, suggesting it learns culture-aware representations that transcend linguistic boundaries.

Table 7: Zero-shot cross-lingual transfer performance (F1 scores). Models trained on source languages, tested on target.

| Model | European | | Asian | | African | |
|---|---|---|---|---|---|---|
| | English→Irish | French→Occitan | Chinese→Tibetan | Hindi→Santali | Swahili→Shona | Arabic→Berber |
| GPT-4 | 42.3 | 48.7 | 38.9 | 35.2 | 31.4 | 33.8 |
| Claude-3 | 40.8 | 46.2 | 37.1 | 33.6 | 29.7 | 32.1 |
| BLOOM-176B | 45.1 | 51.2 | 41.3 | 37.8 | 35.6 | 38.2 |
| Vicuna-13B | 35.7 | 41.3 | 32.4 | 28.9 | 25.1 | 27.6 |
| CulturalLLM | 37.9 | 43.8 | 34.6 | 31.2 | 27.3 | 29.8 |
| ICHLLM-7B | 64.2 | 67.8 | 61.5 | 57.3 | 53.9 | 58.1 |
| ICHLLM-13B | **68.9** | **71.3** | **65.7** | **61.4** | **58.2** | **62.1** |

**Few-shot Learning.** We evaluate few-shot adaptation capabilities on novel ICH domains not seen during training. We select three challenging scenarios: (1) adapting to new UNESCO ICH domains, (2) learning from rare cultural practices, and (3) rapid adaptation to endangered languages with minimal data. ICHLLM demonstrates superior few-shot learning abilities across all scenarios. Even in 0-shot settings, ICHLLM-13B outperforms GPT-4's 10-shot performance on new ICH domains. The model shows particularly strong rapid adaptation for endangered languages, achieving 74.5 F1 with just 10 examples compared to GPT-4's 51.8 F1. This rapid adaptation capability is crucial for ICH preservation, where many practices and languages have limited documented examples. The strong 0-shot performance suggests ICHLLM has learned generalizable cultural representations that transfer effectively to unseen heritage contexts.

**Case Study.** We present qualitative examples to illustrate ICHLLM's superior cultural understanding. Table 9 shows representative outputs from different models on narrative generation and cultural translation tasks.

**Efficiency and Scalability Analysis.** Table 10 compares computational efficiency across models, crucial for practical ICH preservation applications in resource-limited settings. ICHLLM achieves the best performance-per-parameter ratio (10.37 for 7B, 5.92 for 13B), demonstrating efficient cultural

Table 8: Few-shot learning performance (F1 scores) on novel ICH tasks. Models are evaluated with varying numbers of examples.

| Model | New ICH Domain | | | | Rare Cultural Practice | | | | Endangered Language | | | |
|---|---|---|---|---|---|---|---|---|---|---|---|---|
| | 0-shot | 1-shot | 5-shot | 10-shot | 0-shot | 1-shot | 5-shot | 10-shot | 0-shot | 1-shot | 5-shot | 10-shot |
| GPT-4 | 45.2 | 52.3 | 61.7 | 65.4 | 38.9 | 46.2 | 55.8 | 59.3 | 31.5 | 38.7 | 47.2 | 51.8 |
| Claude-3 | 43.8 | 50.9 | 59.2 | 63.1 | 37.4 | 44.8 | 53.6 | 57.9 | 30.2 | 37.1 | 45.8 | 50.3 |
| BLOOM-176B | 32.1 | 38.4 | 45.7 | 49.2 | 25.6 | 31.9 | 38.2 | 42.3 | 22.4 | 27.8 | 34.1 | 38.6 |
| CulturalLLM | 39.7 | 47.2 | 56.1 | 60.8 | 33.2 | 40.5 | 49.7 | 54.1 | 27.9 | 34.6 | 42.8 | 47.5 |
| ICHLLM-7B | 58.3 | 68.9 | 76.2 | 79.8 | 51.7 | 62.4 | 71.3 | 75.6 | 43.2 | 54.8 | 64.7 | 69.3 |
| ICHLLM-13B | **62.7** | **73.5** | **80.9** | **84.2** | **56.3** | **67.8** | **76.4** | **80.1** | **47.9** | **59.6** | **69.8** | **74.5** |

Table 9: Case study examples showing model outputs for ICH tasks. Cultural elements are highlighted.

| Model | Generated Output |
|---|---|
| *Task: Translate the Yoruba concept of "omoluabi" to English* | |
| GPT-4 | Omoluabi means "good person" or "well-behaved individual." |
| ICHLLM-13B | "Omoluabi" encompasses a complete philosophical framework of personhood in Yoruba culture. While often simplified as "good character," it represents someone who embodies ìwà p`ẹl´ẹ (gentle character), ìwà omolúwàbí (integrity), and serves as a cultural exemplar. The concept includes responsibility to community, respect for elders, industriousness, and truthfulness. No single English term captures its depth; it's closer to "a person of complete moral and social excellence who upholds community values." |

knowledge encoding. The 7B model offers optimal deployment efficiency for field applications, maintaining 63 tokens/s inference speed while achieving 72.6% average performance, making it suitable for mobile heritage documentation systems. The training time is comparable to baseline models despite the additional cultural tokens, showing our approach doesn't significantly increase computational overhead.

Table 10: Efficiency metrics for different models. Inference measured on single A100 GPU.

| Model | Parameters | Memory (GB) | Inference (tokens/s) | Training Time | Performance/Param |
|---|---|---|---|---|---|
| GPT-4 | 1.76T | N/A (API) | 20 (API limit) | N/A | 0.037 |
| Claude-3 | N/A | N/A (API) | 30 (API limit) | N/A | N/A |
| BLOOM-176B | 176B | 352 | 8 | N/A | 0.255 |
| Vicuna-13B | 13B | 26 | 42 | 120h | 3.78 |
| Alpaca-7B | 7B | 14 | 68 | 80h | 6.69 |
| CulturalLLM | 7B | 14 | 65 | 100h | 7.60 |
| ICHLLM-7B | 7B | 14 | 63 | 96h | 10.37 |
| ICHLLM-13B | 13B | 26 | 41 | 168h | **5.92** |

**Cultural Fairness Analysis.** We evaluate model fairness across different cultural regions to ensure equitable preservation of diverse heritage. Table 11 shows performance variance across UNESCO regions. ICHLLM demonstrates remarkable fairness with the lowest standard deviation, indicating consistent performance across all cultural regions. While GPT-4 shows a 20.1-point gap between its best and worst regions, ICHLLM-13B maintains only a 4.3-point gap. This equitable performance is crucial for ICH preservation, ensuring that marginalized and underrepresented cultures receive equal support in digital preservation efforts. The balanced representation stems from our careful dataset curation and the use of culture-specific tokens that prevent bias toward dominant cultures.

Table 11: Performance (F1 scores) across different cultural regions, measuring fairness in ICH representation.

| Model | Africa | Arab | Asia-Pacific | Europe | Latin America | North America | Std Dev |
|---|---|---|---|---|---|---|---|
| GPT-4 | 58.3 | 61.7 | 72.8 | 78.4 | 65.2 | 75.6 | 8.21 |
| Claude-3 | 56.9 | 59.4 | 70.2 | 76.8 | 63.7 | 73.9 | 8.07 |
| BLOOM-176B | 42.1 | 48.6 | 51.3 | 53.7 | 45.8 | 49.2 | 4.32 |
| CulturalLLM | 50.3 | 52.8 | 61.4 | 64.9 | 55.7 | 62.1 | 5.89 |
| ICHLLM-7B | 71.8 | 73.2 | 75.9 | 76.4 | 74.1 | 75.3 | 1.89 |
| ICHLLM-13B | 75.4 | 76.8 | 79.2 | 79.7 | 77.6 | 78.9 | **1.68** |

# 7 CONCLUSION

We present MEMORIA, a comprehensive framework for intangible cultural heritage preservation through large language models. The framework consists of three components: (1) ICHLLM, culturally fine-tuned models in 7B and 13B variants that significantly outperform general-purpose LLMs; (2) CHIT, a 159K-sample instruction dataset spanning UNESCO's five ICH domains; and (3) ICHEB, an evaluation benchmark with 6 tasks across 13 datasets introducing novel culture-specific metrics. Our experiments validate MEMORIA's effectiveness in preserving cultural knowledge while maintaining computational efficiency. By open-sourcing all components and establishing standardized benchmarks, we provide the research community with essential infrastructure for advancing AI-driven cultural preservation. Future work will extend coverage to endangered languages and explore multimodal representations for performance-based heritage.

## REPRODUCIBILITY STATEMENT

To ensure full reproducibility of our research, we provide comprehensive documentation across all components of MEMORIA. The complete ICHLLM training pipeline, including hyperparameters, data preprocessing steps, and fine-tuning configurations, is detailed in Section 6 and Appendix H. Our CHIT dataset construction methodology is thoroughly documented in Section 3.1 with step-by-step data collection, filtering, and augmentation procedures. The ICHEB benchmark implementation, including all evaluation metrics and scoring algorithms, is provided in Section 5 with mathematical formulations detailed in Appendix C. Source code for model training, evaluation pipelines, and cultural metrics calculation will be released at publication. All experimental configurations, including computing infrastructure specifications, training schedules, and evaluation protocols, are documented in Appendix H. The novel Cultural Score and Cultural Fidelity metrics are mathematically formalized in Appendix C with complete algorithmic descriptions. Dataset statistics, cultural coverage analysis, and benchmark composition details are provided in Appendix K. Anonymous code and data are currently available at `https://anonymous.4open.science/r/MEMORIA`, with full public release planned upon paper acceptance.

## ETHICS STATEMENT

This research adheres to the ICLR Code of Ethics and addresses several important ethical considerations. Our work focuses exclusively on publicly available intangible cultural heritage materials and respects cultural boundaries by implementing sacred knowledge filters and community consent mechanisms. The CHIT dataset construction process involved extensive consultation with cultural experts and adherence to UNESCO guidelines for ethical heritage documentation. Our Sacred Knowledge Handling (SKH) metric explicitly prevents inappropriate exposure of restricted cultural content, while our evaluation framework includes cultural sensitivity checks to avoid perpetuating stereotypes or misrepresentations. All cultural data sources are properly attributed, and we have established partnerships with heritage communities to ensure respectful representation. The open-source release of MEMORIA is designed to democratize access to cultural preservation tools while maintaining ethical safeguards. This work aims to empower heritage communities rather than extract from them, supporting the UN Declaration on the Rights of Indigenous Peoples and promoting equitable access to digital preservation technologies.

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

## A    THE ANONYMOUS CODE

The anonymous code can be available at `https://anonymous.4open.science/r/ MEMORIA`.

## B    THE USE OF LARGE LANGUAGE MODELS (LLMS)

We leverage a Large Language Model (LLM) to refine and polish the language in the introduction, ensuring clarity and impact.

## C    NOVEL EVALUATION METRICS FOR ICH TASKS

Traditional NLP evaluation metrics fail to capture the unique requirements of intangible cultural heritage preservation, particularly the cultural authenticity, traditional knowledge accuracy, and appropriate representation of indigenous practices. We introduce two novel metrics specifically designed for ICH evaluation tasks.

### C.1    FORMALIZING CULTURAL AUTHENTICITY IN GENERATED NARRATIVES

We propose Cultural Score (CS), a comprehensive metric framework that formalizes the evaluation of narrative authenticity through five empirically-derived dimensions essential for intangible heritage transmission:

**1. Structural Authenticity (SA):** This dimension quantifies conformance to culturally-specific narrative architectures through computational analysis of morphological patterns. Formally defined as:

$$f_{SA}(\mathbf{x}, \mathcal{C}) = \alpha \cdot \text{Align}(\mathbf{x}, \mathcal{T}_c) + \beta \cdot \text{Struct}(\mathbf{x}, \mathcal{P}_c) + \gamma \cdot \text{Form}(\mathbf{x}, \mathcal{F}_c)$$

where:

- $\text{Align}(\mathbf{x}, \mathcal{T}_c)$ measures sequence alignment with cultural templates $\mathcal{T}_c$ using modified Needleman-Wunsch algorithm with culture-specific gap penalties
- $\text{Struct}(\mathbf{x}, \mathcal{P}_c)$ evaluates adherence to narrative patterns $\mathcal{P}_c$ through graph-based story structure analysis
- $\text{Form}(\mathbf{x}, \mathcal{F}_c)$ assesses formulaic expression density using n-gram matching against cultural corpora $\mathcal{F}_c$
- $\alpha = 0.4, \beta = 0.35, \gamma = 0.25$ are empirically determined weights

**2. Motif Fidelity (MF):** This component evaluates the semantic integrity of cultural motifs through ontological mapping and symbolic analysis. Mathematically formalized as:

$$f_{MF}(\mathbf{x}, \mathcal{O}_c) = \frac{1}{|\mathcal{M}|} \sum_{m \in \mathcal{M}} \text{sim}(e_m, \mathcal{O}_c) \cdot \text{freq}(m, \mathbf{x}) \cdot \text{context}(m, \mathbf{x})$$

where:

- $\mathcal{M}$ denotes the set of detected motifs in text $\mathbf{x}$ using ontology-guided NER
- $\text{sim}(e_m, \mathcal{O}_c)$ measures semantic similarity between entity $e_m$ and cultural ontology $\mathcal{O}_c$ using knowledge graph embeddings
- $\text{freq}(m, \mathbf{x})$ quantifies motif frequency normalized by text length
- $\text{context}(m, \mathbf{x})$ evaluates contextual appropriateness using BERT-based cultural context classifiers

**3. Linguistic Authenticity (LA):** This metric dimension captures prosodic and stylistic features through computational stylometry and discourse analysis. Defined as:

$$f_{LA}(\mathbf{x}, \mathcal{L}_c) = \lambda_1 \cdot \text{Pros}(\mathbf{x}, \mathcal{R}_c) + \lambda_2 \cdot \text{Style}(\mathbf{x}, \mathcal{S}_c) + \lambda_3 \cdot \text{Dist}(\mathbf{x}, \mathcal{D}_c)$$

where:

- $\text{Pros}(\mathbf{x}, \mathcal{R}_c) = \frac{1}{N} \sum_{i=1}^{N} \text{RhythmSim}(s_i, \mathcal{R}_c)$ measures prosodic patterns against cultural rhythm templates $\mathcal{R}_c$
- $\text{Style}(\mathbf{x}, \mathcal{S}_c) = \text{KL}(\mathcal{P}_x || \mathcal{P}_c)^{-1}$ evaluates stylistic similarity using KL divergence between n-gram distributions
- $\text{Dist}(\mathbf{x}, \mathcal{D}_c) = \cos(\text{BERT}(\mathbf{x}), \text{BERT}(\mathcal{D}_c))$ measures distributional semantic alignment with cultural discourse models
- $\lambda_1 = 0.3, \lambda_2 = 0.4, \lambda_3 = 0.3$ are linguistically motivated weights

**4. Value Alignment (VA):** This evaluative component quantifies axiological coherence between generated content and indigenous epistemological frameworks. Formally:

$$f_{VA}(\mathbf{x}, \mathcal{V}_c) = \text{sigmoid}\left( \sum_{v \in \mathcal{V}_c} w_v \cdot \phi_v(\mathbf{x}) - \sum_{t \in \mathcal{T}_c} \tau_t \cdot \psi_t(\mathbf{x}) \right)$$

where:

- $\phi_v(\mathbf{x})$ detects value indicator $v$ using multi-label classification with attention mechanisms
- $\mathcal{V}_c = \{v_1, ..., v_k\}$ represents the cultural value system with learned weights $w_v$
- $\psi_t(\mathbf{x})$ identifies taboo violation $t$ using adversarial training for robust detection
- $\mathcal{T}_c = \{t_1, ..., t_m\}$ denotes cultural transgression patterns with penalty weights $\tau_t$
- sigmoid ensures output normalization to $[0, 1]$ range

**5. Transmission Appropriateness (TA):** This pedagogical dimension evaluates narratives' efficacy as vehicles for intergenerational knowledge transfer. Computed as:

$$f_{TA}(\mathbf{x}, \mathcal{A}) = \frac{1}{|\mathcal{A}|} \sum_{a \in \mathcal{A}} \text{Pedagogy}(\mathbf{x}, a) \cdot \text{Clarity}(\mathbf{x}, a) \cdot \text{Relevance}(\mathbf{x}, t)$$

where:

- $\text{Pedagogy}(\mathbf{x}, a) = 1 - \frac{|\text{FRE}(\mathbf{x}) - \text{FRE}_{\text{target}}(a)|}{100}$ measures age-appropriate complexity using Flesch Reading Ease adapted for cultural contexts
- $\text{Clarity}(\mathbf{x}, a) = \frac{\text{count(explicit\_knowledge)}}{\text{count(implicit\_knowledge)} + \epsilon}$ quantifies knowledge explicitness ratio
- $\text{Relevance}(\mathbf{x}, t) = \exp(-\beta \cdot d_{semantic}(\mathbf{x}, \mathcal{C}_t))$ evaluates temporal relevance with decay factor $\beta = 0.1$
- $\mathcal{A}$ denotes target audience segments with demographic parameters

The composite Cultural Score is formally defined as a weighted linear combination:

$$CS = \sum_{i \in \{SA, MF, LA, VA, TA\}} w_i \cdot f_i(\mathbf{x})$$

where $w_i$ represents empirically-derived weighting coefficients obtained through maximum likelihood estimation on expert-annotated corpora, with baseline values: $w_{SA} = 0.25, w_{MF} = 0.20, w_{LA} = 0.20, w_{VA} = 0.20, w_{TA} = 0.15$.

---

**Algorithm 1** Cultural Score Computation Pipeline

---

1: **Input:** Text $\mathbf{x}$, Cultural context $\mathcal{C} = \{\mathcal{T}_c, \mathcal{O}_c, \mathcal{L}_c, \mathcal{V}_c, \mathcal{A}\}$
2: **Output:** Cultural Score $CS \in [0, 1]$
3: **// Phase 1: Structural Analysis**
4: $templates \leftarrow \text{LoadCulturalTemplates}(\mathcal{T}_c)$
5: $sa\_score \leftarrow \text{ComputeStructuralAuth}(\mathbf{x}, templates)$
6: **// Phase 2: Motif Detection**
7: $motifs \leftarrow \text{ExtractMotifs}(\mathbf{x}, \mathcal{O}_c)$
8: $mf\_score \leftarrow \text{EvaluateMotifFidelity}(motifs, \mathcal{O}_c)$
9: **// Phase 3: Linguistic Analysis**
10: $prosody \leftarrow \text{ExtractProsodicFeatures}(\mathbf{x})$
11: $style \leftarrow \text{ComputeStyleVector}(\mathbf{x})$
12: $la\_score \leftarrow \text{MeasureLinguisticAuth}(prosody, style, \mathcal{L}_c)$
13: **// Phase 4: Value Assessment**
14: $values \leftarrow \text{DetectValueIndicators}(\mathbf{x}, \mathcal{V}_c)$
15: $violations \leftarrow \text{IdentifyTaboos}(\mathbf{x}, \mathcal{T}_c)$
16: $va\_score \leftarrow \text{ComputeValueAlignment}(values, violations)$
17: **// Phase 5: Transmission Evaluation**
18: $pedagogy \leftarrow \text{AssessPedagogicalValue}(\mathbf{x}, \mathcal{A})$
19: $ta\_score \leftarrow \text{EvaluateTransmission}(pedagogy)$
20: **// Phase 6: Score Aggregation**
21: $CS \leftarrow 0.25 \cdot sa\_score + 0.20 \cdot mf\_score + 0.20 \cdot la\_score$
22: $\qquad + 0.20 \cdot va\_score + 0.15 \cdot ta\_score$
23: **return** $CS$

---

Each dimensional score $f_i(\mathbf{x}) \in [0, 1]$ is derived through the computational framework detailed above. To enable large-scale deployment, we develop a neural scoring function based on fine-tuned BERT architectures, achieving $\rho = 0.82$ Pearson correlation coefficient with expert annotations on a held-out validation corpus ($n = 2,847$, $p < 0.001$).

**Implementation Framework:** Algorithm 1 presents the computational pipeline for Cultural Score calculation, integrating the five dimensional metrics through a hierarchical evaluation architecture.

C.2 QUANTIFYING SEMANTIC PRESERVATION IN CROSS-CULTURAL TRANSFER

We introduce Cultural Fidelity (CF), a novel metric that quantifies the preservation of culturally-embedded semantics during cross-lingual transfer, extending beyond surface-level linguistic correspondence to capture deep cultural meaning through:

**1. Conceptual Preservation (CP):** This dimension quantifies the semantic integrity of culture-bound concepts through lexical-semantic analysis and frame semantics. Formally defined as:

$$f_{CP}(\mathbf{s}, \mathbf{t}) = \mu_1 \cdot \text{Lex}(\mathbf{s}, \mathbf{t}) + \mu_2 \cdot \text{Frame}(\mathbf{s}, \mathbf{t}) + \mu_3 \cdot \text{Role}(\mathbf{s}, \mathbf{t})$$

where:

- $\text{Lex}(\mathbf{s}, \mathbf{t}) = 1 - \frac{|\mathcal{L}_{\text{gap}}|}{|\mathcal{L}_{\text{total}}|}$ quantifies lexical gap ratio, where $\mathcal{L}_{\text{gap}}$ denotes untranslatable concepts requiring periphrastic expansion

- $\text{Frame}(\mathbf{s}, \mathbf{t}) = \frac{1}{|\mathcal{F}|} \sum_{f \in \mathcal{F}} \max_{f' \in \mathcal{F}'} \text{sim}_{\text{frame}}(f, f')$ measures frame semantic alignment using FrameNet embeddings

- $\text{Role}(\mathbf{s}, \mathbf{t}) = \prod_{r \in \mathcal{R}} P(r_t | r_s, \mathcal{C})^{1/|\mathcal{R}|}$ evaluates socio-cultural role preservation through conditional probability

- $\mu_1 = 0.35, \mu_2 = 0.35, \mu_3 = 0.30$ are linguistically motivated weights

**2. Metaphorical Mapping (MM):** This component assesses the transfer of figurative language through cognitive linguistic analysis and conceptual metaphor theory. Mathematically expressed as:

$$f_{MM}(\mathbf{s}, \mathbf{t}) = \frac{1}{|\mathcal{M}_s|} \sum_{m_s \in \mathcal{M}_s} \max_{m_t \in \mathcal{M}_t} \text{MetScore}(m_s, m_t, \mathcal{C}_s, \mathcal{C}_t)$$

where:

$$\text{MetScore}(m_s, m_t, \mathcal{C}_s, \mathcal{C}_t) = \eta_1 \cdot \text{Dom}(m_s, m_t) + \eta_2 \cdot \text{Func}(m_s, m_t) + \eta_3 \cdot \text{Expl}(m_t)$$

- $\text{Dom}(m_s, m_t) = \cos(\text{CMT}_s(m_s), \text{CMT}_t(m_t))$ measures conceptual domain alignment using Conceptual Metaphor Theory embeddings

- $\text{Func}(m_s, m_t) = \frac{\text{effect}(m_s, \mathcal{C}_s) \cap \text{effect}(m_t, \mathcal{C}_t)}{|\text{effect}(m_s, \mathcal{C}_s)|}$ evaluates functional equivalence preservation

- $\text{Expl}(m_t) = \begin{cases} 1.0 & \text{if metaphor is transparent} \\ 0.7 + 0.3 \cdot P(\text{meta-comment}) & \text{if requires explanation} \end{cases}$

- $\eta_1 = 0.4, \eta_2 = 0.35, \eta_3 = 0.25$ are empirically derived coefficients

**3. Pragmatic Accuracy (PA):** This metric evaluates the preservation of illocutionary force and socio-pragmatic appropriateness through discourse analysis methodologies. Formalized as:

$$f_{PA}(\mathbf{s}, \mathbf{t}) = \text{sigmoid}\left(\sum_{i=1}^{N} \theta_i \cdot \phi_i(\mathbf{s}, \mathbf{t})\right)$$

where pragmatic features $\phi_i$ include:

- $\phi_1(\mathbf{s}, \mathbf{t}) = 1 - \frac{|FTA_s - FTA_t|}{\max(FTA_s, FTA_t) + \epsilon}$ measures Face Threatening Act (FTA) weight preservation using Brown-Levinson model

- $\phi_2(\mathbf{s}, \mathbf{t}) = \frac{1}{|\mathcal{U}|} \sum_{u \in \mathcal{U}} P(I_t = I_s | u, \mathcal{C})$ evaluates illocutionary force alignment through contextual inference

- $\phi_3(\mathbf{s}, \mathbf{t}) = \exp\left(-\sum_{m=1}^{4} |g_m^s - g_m^t|\right)$ quantifies Gricean maxim adherence across quantity, quality, relation, manner

- $\phi_4(\mathbf{s}, \mathbf{t}) = \text{JSD}(\mathcal{P}_{\text{reg}}^s || \mathcal{P}_{\text{reg}}^t)^{-1}$ measures register preservation using Jensen-Shannon divergence

- $\theta = [0.3, 0.25, 0.25, 0.2]$ are discourse-theoretically motivated weights

**4. Sacred Knowledge Handling (SKH):** This dimension assesses the ethical treatment of epistemologically-restricted content through culturally-informed access control mechanisms. Computed as:

$$f_{SKH}(\mathbf{s}, \mathbf{t}, \mathcal{A}) = \prod_{k \in \mathcal{K}_{\text{sacred}}} \text{Protect}(k, \mathbf{t}, \mathcal{A})$$

where:

$$\text{Protect}(k, \mathbf{t}, \mathcal{A}) = \begin{cases} 1.0 & \text{if } \rho(k) = 0 \text{ (public knowledge)} \\ \sigma(\alpha \cdot \text{Gen}(k, \mathbf{t}) - \beta \cdot \text{Spec}(k, \mathbf{t})) & \text{if } 0 < \rho(k) < 3 \\ \mathbb{I}[\text{absent}(k, \mathbf{t})] & \text{if } \rho(k) = 3 \text{ (forbidden)} \end{cases}$$

- $\rho(k) \in \{0, 1, 2, 3\}$ denotes restriction level from public to forbidden, determined by cultural authority matrices

- $\text{Gen}(k, \mathbf{t}) = 1 - \frac{d_{\text{semantic}}(k_{\text{original}}, k_{\text{translated}})}{d_{\text{max}}}$ measures appropriate generalization

- $\text{Spec}(k, \mathbf{t}) = \frac{|\text{details}(k, \mathbf{t})|}{|\text{details}(k, \mathbf{s})|}$ quantifies specificity preservation

- $\mathbb{I}[\cdot]$ is the indicator function, $\alpha = 0.6, \beta = 0.4$ balance generalization vs. information loss

---

**Algorithm 2** Cultural Fidelity Computation Pipeline

---

1: **Input:** Source text $\mathbf{s}$, Target text $\mathbf{t}$, Cultural contexts $\mathcal{C}_s, \mathcal{C}_t$, Audience $\mathcal{A}$
2: **Output:** Cultural Fidelity $CF \in [0, 1]$
3: **// Phase 1: Conceptual Analysis**
4: $gaps \leftarrow$ DetectLexicalGaps($\mathbf{s}, \mathbf{t}, \mathcal{C}_s, \mathcal{C}_t$)
5: $frames \leftarrow$ ExtractFrameSemantics($\mathbf{s}, \mathbf{t}$)
6: $roles \leftarrow$ AnalyzeCulturalRoles($\mathbf{s}, \mathbf{t}, \mathcal{C}_s$)
7: $cp\_score \leftarrow$ ComputeConceptualPreservation($gaps, frames, roles$)
8: **// Phase 2: Metaphor Processing**
9: $metaphors_s \leftarrow$ ExtractMetaphors($\mathbf{s}, \mathcal{C}_s$)
10: $metaphors_t \leftarrow$ MapMetaphors($metaphors_s, \mathcal{C}_t$)
11: $mm\_score \leftarrow$ EvaluateMetaphoricalMapping($metaphors_s, metaphors_t$)
12: **// Phase 3: Pragmatic Analysis**
13: $fta_s, fta_t \leftarrow$ ComputeFTAWeights($\mathbf{s}, \mathbf{t}$)
14: $force_s, force_t \leftarrow$ ExtractIllocutionaryForce($\mathbf{s}, \mathbf{t}$)
15: $maxims \leftarrow$ CheckGriceanMaxims($\mathbf{s}, \mathbf{t}$)
16: $pa\_score \leftarrow$ AssessPragmaticAccuracy($fta_s, fta_t, force_s, force_t, maxims$)
17: **// Phase 4: Sacred Content Protection**
18: $sacred \leftarrow$ IdentifySacredContent($\mathbf{s}, \mathcal{C}_s$)
19: **for each** $k \in sacred$ **do**
20:  $restriction \leftarrow$ GetRestrictionLevel($k, \mathcal{C}_s$)
21:  $protection \leftarrow$ VerifyProtection($k, \mathbf{t}, restriction, \mathcal{A}$)
22: **end for**
23: $skh\_score \leftarrow$ AggregateProtectionScores($protection$)
24: **// Phase 5: Score Aggregation**
25: $CF \leftarrow 0.30 \cdot cp\_score + 0.25 \cdot mm\_score$
26:   $+ 0.25 \cdot pa\_score + 0.20 \cdot skh\_score$
27: **return** $CF$

---

- $\mathcal{A}$ represents audience demographic parameters for access control verification

The composite Cultural Fidelity metric is formally defined through the weighted aggregation function:

$$CF(\mathbf{s}, \mathbf{t}) = \sum_{j \in \{CP, MM, PA, SKH\}} \omega_j \cdot g_j(\mathbf{s}, \mathbf{t})$$

where $\mathbf{s}$ and $\mathbf{t}$ denote source and target texts respectively, and weighting parameters $\omega_{CP} = 0.30, \omega_{MM} = 0.25, \omega_{PA} = 0.25, \omega_{SKH} = 0.20$ are derived through conjoint analysis with indigenous knowledge holders ($n = 127$).

**Implementation Framework:** Algorithm 2 details the computational pipeline for Cultural Fidelity evaluation in cross-cultural translation tasks.

## C.3 Metric Validation and Implementation Details

This section provides empirical validation results and comprehensive technical specifications for reproducing the Cultural Score and Cultural Fidelity metrics.

### C.3.1 Empirical Validation

**Inter-rater Reliability Analysis:** We conduct multi-scale validation to establish metric reliability. Initial assessment with five senior cultural heritage experts (minimum 15 years experience, representing African, Asian, European, American Indigenous, and Oceanic traditions) yields Krippendorff's $\alpha = 0.79$ (95% CI: [0.76, 0.82]) for Cultural Score and $\alpha = 0.81$ (95% CI: [0.78, 0.84]) for Cultural Fidelity. Extended validation with 28 regional experts across 14 countries maintains high agreement ($\alpha = 0.77$ for CS, $\alpha = 0.79$ for CF), confirming cross-cultural stability.

**Convergent Validity:** Neural scoring functions achieve strong correlation with expert judgments across three validation tiers:

- **Expert Validation** (n=2,847 samples, 28 experts): CS: $\rho = 0.82$, CF: $\rho = 0.78$
- **Crowd-sourced Validation** (n=10,234 samples, 412 cultural practitioners): CS: $\rho = 0.76$, CF: $\rho = 0.74$
- **Community Validation** (n=847 samples, 12 indigenous communities): CS: $\rho = 0.79$, CF: $\rho = 0.77$

The consistency across validation scales (expert-crowd correlation: $r = 0.84$, p<0.001) confirms that our metrics capture genuine cultural dimensions rather than expert-specific biases.

**Ecological Validity:** Field validation across 12 indigenous communities ($n = 847$ participants) demonstrates strong predictive validity for real-world heritage outcomes. Content achieving threshold scores (CS/CF > 0.7) exhibits significantly higher adoption rates for pedagogical applications (OR = 3.2, $\chi^2 = 47.3$, $p < 0.001$) and elevated authenticity ratings on culturally-grounded assessment instruments (Cohen's $d = 1.47$, $p < 0.001$).

**Comparison with Alternative Metrics:** We validate our metrics against existing cultural assessment approaches:

- **vs. Human Judgment Baseline**: CS/CF predictions align with majority human vote in 83.7% of cases (chance: 20%)
- **vs. Hofstede Cultural Dimensions**: Moderate correlation ($r = 0.43$) as expected, since Hofstede focuses on contemporary rather than heritage culture
- **vs. BERTScore Alone**: Adding CS/CF improves heritage content ranking by 31.2% (NDCG@10: 0.74 vs 0.51)
- **vs. Domain Expert Holistic Scoring**: Strong correlation ($r = 0.86$) but our metrics provide interpretable dimensional breakdowns

### C.3.2 IMPLEMENTATION DETAILS

To enable practical deployment of Cultural Score and Cultural Fidelity metrics, we construct a comprehensive cultural knowledge base spanning 147 traditions across five UNESCO domains. This knowledge base comprises 12,847 hierarchically-organized cultural concept nodes, 45,672 narrative templates extracted from 89 language families, and 3,456 cross-cultural metaphorical mappings validated by domain experts. These resources enable the metrics to recognize culture-specific patterns, narrative structures, and conceptual mappings essential for authentic heritage assessment.

The neural scoring model employs a 12-layer transformer encoder (hidden_dim=768, attention_heads=12) augmented with a 4-layer cultural attention mechanism (cultural_emb_dim=512) that learns culture-specific representations. The model outputs scores through a 5-head multi-task architecture corresponding to the five CS dimensions. Training uses AdamW optimizer with learning rate 5e-5 and cosine annealing, batch size 32 with gradient accumulation, and early stopping based on validation performance. For efficient deployment, we implement INT8 quantization (0.3% accuracy loss) and pre-computed cultural embeddings, achieving 47ms average inference latency on V100 GPUs while maintaining scoring quality.

The structural alignment component for narrative assessment adapts the Needleman-Wunsch algorithm with culture-specific parameters learned from 10,000 aligned narrative pairs. This enables recognition of both linear Western narrative structures and cyclical patterns common in oral traditions, with differential gap penalties and substitution matrices tailored to each cultural tradition's storytelling conventions.

## D TRADITIONAL METHODS FOR ICH PRESERVATION

Traditional ICH preservation methods have progressed through three major paradigms, each revealing critical limitations that MEMORIA addresses. Early digitization efforts (1990s) using static repositories like CDWA provided structured metadata but failed to capture the dynamic, performative nature of living heritage, with single ceremonial practices requiring weeks of manual documentation. Knowledge graph technologies (2010s) enabled modeling complex cultural relationships through projects like Europeana, yet remained constrained by Western-biased ontologies that misrepresented

indigenous knowledge systems and required prohibitive computational resources. UNESCO's standardized protocols, while ensuring consistency, demand 50-100 pages of technical documentation per element, creating barriers that exclude 78% of communities lacking institutional support from global heritage discourse. Table 12 demonstrates MEMORIA's advantages across operational dimensions. Our framework achieves 100× throughput improvement over manual methods by processing 159K cultural narratives in parallel, while enabling full interactivity through both analytical queries and generative tasks. Most critically, MEMORIA's native cross-lingual capabilities democratize heritage preservation, allowing communities to document traditions in indigenous languages without requiring translation into dominant lingua francas—addressing the fundamental accessibility barriers that have historically excluded marginalized communities from digital preservation initiatives.

Table 12: Comparative analysis of ICH preservation methods across key operational dimensions

| Method | Scalability | Interactivity | Cross-lingual | Generation |
|---|---|---|---|---|
| Digital Archives (CDWA) | Low | None | Limited | No |
| Knowledge Graphs | Medium | Query-only | No | No |
| UNESCO Documentation | Low | None | Manual | No |
| Template Systems | Medium | Limited | No | Template-only |
| Rule-based NLP | Medium | Limited | Partial | No |
| **MEMORIA (Ours)** | **High** | **Full** | **Yes** | **Yes** |

# E  PERFORMANCE ACROSS DIFFERENT MODEL ARCHITECTURES

Table 13 presents comprehensive architectural comparisons revealing decoder-only models' superiority for ICH tasks. Encoder-only models (BERT, XLM-RoBERTa) achieve only 42.3-48.7% F1 on entity recognition while lacking essential generative capabilities for narrative preservation. Encoder-decoder architectures (mT5 family) offer intermediate performance but suffer from cultural homogenization in longer generations and computational inefficiency for resource-constrained institutions. Decoder-only models show clear performance scaling from LLaMA-7B (63.1%) to GPT-3 (65.0%), yet our culturally-tuned ICHLLM achieves superior results (71.1-74.7%) with 25× fewer parameters through targeted instruction tuning, with particularly strong gains in culture-specific tasks like Heritage-Trans (+8.7 points) and Story-Gen (+8.3 points). Memory requirements critically influence deployment feasibility: while GPT-3's API-based approach raises data sovereignty concerns for indigenous communities, ICHLLM's single-GPU compatibility (14GB for 7B model) enables local deployment, ensuring communities maintain control over their cultural knowledge while achieving state-of-the-art performance. This architectural efficiency democratizes access to advanced ICH preservation technology, particularly crucial for the 73% of cultural institutions with limited computational resources.

Table 13: Detailed performance comparison across model architectures on ICHEB benchmark

| Model | Size | Heritage-NER | Culture-QA | Story-Gen | Heritage-Trans | Avg | Memory (GB) |
|---|---|---|---|---|---|---|---|
| *Encoder-only Models* | | | | | | | |
| BERT-base | 110M | 42.3 | 38.1 | - | 45.2 | 41.9 | 0.4 |
| XLM-RoBERTa | 550M | 48.7 | 42.5 | - | 52.8 | 48.0 | 2.1 |
| *Encoder-Decoder Models* | | | | | | | |
| mT5-base | 580M | 51.2 | 45.3 | 58.4 | 61.7 | 54.2 | 2.3 |
| mT5-large | 1.2B | 54.8 | 48.9 | 62.1 | 64.3 | 57.5 | 4.7 |
| *Decoder-only Models* | | | | | | | |
| GPT-3 | 175B | 58.3 | 61.2 | 68.5 | 72.1 | 65.0 | API |
| LLaMA-7B | 7B | 56.7 | 59.4 | 65.3 | 70.8 | 63.1 | 13.5 |
| LLaMA-13B | 13B | 59.2 | 62.8 | 67.9 | 73.4 | 65.8 | 26.0 |
| Alpaca-7B | 7B | 57.1 | 60.2 | 66.1 | 71.2 | 63.7 | 13.5 |
| Vicuna-13B | 13B | 60.3 | 63.5 | 68.7 | 74.1 | 66.7 | 26.0 |
| ICHLLM-7B | 7B | 64.5 | 68.3 | 72.8 | 78.9 | 71.1 | 14.0 |
| ICHLLM-13B | 13B | **68.4** | **71.9** | **76.2** | **82.1** | **74.7** | 26.0 |

## F    DATASET STATISTICS AND CULTURAL DISTRIBUTION

CHIT's construction involved 18-month coordination across 147 cultural institutions, 89 indigenous communities, and 23 UNESCO offices, prioritizing authentic community voices (62% from tradition bearers) over academic interpretations. This required developing new consent protocols for collective ownership contexts—a critical ethical innovation absent from standard NLP practices. The geographic distribution reflects persistent digital divides: Asia-Pacific dominates (37.1%) due to systematic government digitization efforts, while Africa's underrepresentation (17% despite 30% of global linguistic diversity) reveals infrastructure gaps, with 43% of African content from just three countries. Language family distribution similarly shows Indo-European overrepresentation (33% of dataset, 6% of global languages), while 15,800 minority language samples required custom tokenization for non-standardized orthographies, including newly developed writing systems that raise fundamental questions about technological mediation of cultural authenticity. Table 14 reveals UNESCO domain interconnectedness, with oral traditions (30%) foundational to all categories. Critically, 67% of samples span multiple domains—traditional crafts (17%) invariably encompass oral instructions, ritualistic practices (21%), and ecological knowledge (9%)—necessitating sophisticated multi-label schemes that transcend UNESCO's discrete categorization to capture living heritage's holistic nature. This multidimensionality fundamentally shaped our training strategy, requiring models to understand cultural practices as integrated systems rather than isolated categories.

Table 14: CHIT dataset statistics across cultural and linguistic dimensions

| Dimension | Category | Count (%) |
|---|---|---|
| Geographic | Asia-Pacific | 58,640 (37.1%) |
| | Europe | 34,760 (22.0%) |
| | Africa | 26,860 (17.0%) |
| | Americas | 22,140 (14.0%) |
| | Arab States | 15,600 (9.9%) |
| UNESCO Domain | Oral traditions | 47,400 (30.0%) |
| | Performing arts | 36,340 (23.0%) |
| | Social practices | 33,180 (21.0%) |
| | Traditional crafts | 26,860 (17.0%) |
| | Nature knowledge | 14,220 (9.0%) |
| Language Family | Indo-European | 52,140 (33.0%) |
| | Sino-Tibetan | 34,760 (22.0%) |
| | Niger-Congo | 20,540 (13.0%) |
| | Austronesian | 18,960 (12.0%) |
| | Afro-Asiatic | 15,800 (10.0%) |
| | Others | 15,800 (10.0%) |

## G    DATASET SPLITS AND DATA CONTAMINATION PREVENTION

### G.1    TRAIN/VALIDATION/TEST SPLIT METHODOLOGY

The creation of rigorous data splits for CHIT required addressing unique challenges in cultural heritage data, where simple random splitting could introduce severe biases or data leakage. Our multi-stage stratified splitting protocol ensures both statistical validity and cultural representativeness across all evaluation dimensions.

**Split Ratios and Distribution:** From the initial 178,285 instruction samples, we first apply sacred knowledge filtering and quality control, reducing to 159,000 samples for model development. These are divided into:

- **Training Set**: 127,200 samples (80% of 159K)
- **Validation Set**: 15,900 samples (10% of 159K)
- **Test Set**: 15,900 samples (10% of 159K)

The validation and test sets shown in Table 3 represent task-specific subsets sampled from these held-out partitions, carefully balanced to maintain cultural and linguistic diversity while enabling focused evaluation on each capability.

**Stratified Sampling Protocol:** To prevent systematic biases, we implement hierarchical stratified sampling across four dimensions:

1. **Cultural Region**: Each UNESCO region (Asia-Pacific, Europe, Africa, Americas, Arab States) maintains its proportional representation ±1% across all splits

2. **Task Type**: The six task categories preserve their original distribution ratios within 0.5% tolerance

3. **Language Family**: Major language families maintain representation above minimum thresholds (100 samples per split for families with >1000 total samples)

4. **Temporal Source**: Historical periods (pre-1900, 1900-1950, 1950-2000, post-2000) are balanced to prevent temporal bias

**Cultural Integrity Constraints:** Beyond statistical stratification, we enforce cultural coherence rules:

- **Narrative Completeness**: Complete stories, songs, or ritual descriptions remain within single splits (no fragmenting cultural narratives)

- **Community Clustering**: All content from a specific indigenous community stays in the same split to respect collective ownership

- **Lineage Preservation**: Related practices from the same cultural lineage remain together

- **Sacred Knowledge Isolation**: Filtered sensitive content is completely excluded from all splits, not redistributed

G.2   DATA CONTAMINATION PREVENTION AND VERIFICATION

Preventing test set contamination in cultural heritage data presents unique challenges due to oral traditions' multiple recorded versions and cross-cultural knowledge diffusion. Our comprehensive contamination prevention protocol employs both automated and expert-validated methods.

**Multi-Level Deduplication Pipeline:**

1. **Exact Match Removal**: SHA-256 hashing identifies identical texts across splits (removed: 3,247 duplicates)

2. **Near-Duplicate Detection**: MinHash with Jaccard similarity threshold 0.85 catches paraphrases (removed: 7,893 near-duplicates)

3. **Semantic Similarity Filtering**: SBERT embeddings with cosine similarity >0.95 identify semantic duplicates across languages (removed: 4,156 samples)

4. **Cultural Variant Consolidation**: Expert review identifies variant tellings of same narratives, assigning all versions to single split (consolidated: 2,341 narrative clusters)

**Cross-Contamination Verification:** We implement rigorous checks against external data sources:

- **Pre-training Corpus Overlap**: Compare against Common Crawl, Wikipedia, and other LLaMA pre-training sources using 13-gram overlap detection
  - Results: <0.3% overlap with Common Crawl (primarily UNESCO public descriptions)
  - Wikipedia overlap: 1.7% (mostly factual cultural information, retained as legitimate knowledge)

- **Benchmark Contamination**: Check against 47 existing NLP benchmarks and cultural datasets
  - Zero overlap with standard NLP benchmarks (GLUE, SuperGLUE, etc.)
  - 2.1% overlap with CulturalBench (removed from test set)

- **Model Output Verification**: Generate outputs from pre-trained LLaMA to detect potential memorization
  - Perplexity analysis shows no anomalous memorization patterns
  - Manual inspection of 1,000 random samples finds no verbatim reproduction

**Temporal and Source Isolation:** To prevent indirect contamination:

- **Temporal Cutoff**: Test set prioritizes recently documented traditions (post-2020) unlikely to appear in pre-training data
- **Source Exclusivity**: 30% of test data comes from partnerships with communities that have never previously digitized their heritage
- **Language Isolation**: 15% of test samples are in languages with <10,000 digital documents worldwide

### G.3 Statistical Validation of Split Quality

We validate split quality through comprehensive statistical tests:

Table 15: Statistical validation of train/validation/test splits across key dimensions

| Test Metric | Chi-square | p-value | Result |
| --- | --- | --- | --- |
| Cultural region distribution | $\chi^2(8)=0.43$ | 0.999 | No significant difference |
| Task type distribution | $\chi^2(10)=0.21$ | 0.999 | No significant difference |
| Language family distribution | $\chi^2(10)=0.67$ | 0.999 | No significant difference |
| Narrative length distribution | KS=0.018 | 0.973 | No significant difference |
| Complexity score distribution | KS=0.021 | 0.946 | No significant difference |

**Information Leakage Analysis:** We employ mutual information analysis to detect subtle leakage:

- Mutual information between train and test features: I(X_train; X_test) = 0.0043 bits (negligible)
- Conditional entropy analysis: H(Y_test|X_train) = $0.9987 \times$ H(Y_test) (near-maximum uncertainty)
- Feature importance stability: Pearson correlation of feature importance between splits = 0.982 (high stability)

This comprehensive approach ensures that our evaluation results reflect genuine model capabilities rather than data memorization or systematic biases, addressing a critical concern in large-scale language model evaluation.

## H  Training Details and Hyperparameters

Table 16 summarizes optimal configurations from 127 systematic experiments balancing cultural knowledge acquisition with computational efficiency. Differential learning rates (2e-5 for 7B, 1e-5 for 13B) prevent catastrophic forgetting while enabling deep cultural integration, with larger models requiring gentler optimization to maintain stability on specialized domains. LoRA (rank 64, alpha 16) provides optimal parameter efficiency—full fine-tuning yielded only 1.2% improvement while requiring 8× memory and introducing low-resource language instabilities. Gradient accumulation strategies (4 steps for 7B, 8 steps for 13B) achieve 512 effective batch size on single-node clusters, critical for the 73% of cultural institutions with ≤4 GPUs, while 2048 token length covers 94% of CHIT samples without truncation. The bf16 precision choice exemplifies our accessibility-focused design: despite fp32's marginally better stability, no significant performance differences emerged (p>0.05), while bf16's 2× memory savings enables training on commercial GPUs, democratizing advanced ICH preservation. Three-epoch training with early stopping ensures convergence across all cultural domains while preventing overfitting, with particular attention to low-resource domains through stratified sampling to maintain representational equity in the final model.

Table 16: Training configurations for ICHLLM models

| Hyperparameter | ICHLLM-7B | ICHLLM-13B |
|---|---|---|
| Base Model | LLaMA-7B | LLaMA-13B |
| Learning Rate | 2e-5 | 1e-5 |
| Batch Size | 128 | 64 |
| Gradient Accumulation | 4 | 8 |
| Max Sequence Length | 2048 | 2048 |
| Training Epochs | 3 | 3 |
| Warmup Steps | 500 | 500 |
| Weight Decay | 0.01 | 0.01 |
| Optimizer | AdamW | AdamW |
| Precision | bf16 | bf16 |
| LoRA Rank | 64 | 64 |
| LoRA Alpha | 16 | 16 |
| Dropout | 0.1 | 0.1 |

# I   DETAILED CASE STUDIES

This appendix presents comprehensive case studies demonstrating ICHLLM's capabilities across diverse ICH tasks. Each case includes comparative analysis with baseline models, highlighting specific improvements in cultural understanding, sensitivity, and accuracy.

## I.1   ENTITY RECOGNITION IN MULTI-CULTURAL CONTEXTS

Entity recognition in ICH contexts requires understanding not just linguistic patterns but also deep cultural semantics. Traditional NER systems often fail to identify culture-specific roles, sacred objects, and ritual contexts. Our analysis shows that ICHLLM achieves 64.5% F1 score on Heritage-NER tasks, significantly outperforming GPT-4's 58.3%, primarily due to its training on culturally-annotated datasets that distinguish between generic and culture-specific entity types. Table 17 illustrates this capability through a Javanese shadow puppetry example.

Table 17: Case Study: Cultural Entity Recognition in Indonesian Wayang Kulit

| | |
|---|---|
| **Input Text** | "During the *ruwatan* ceremony, the *dalang* performs *Murwakala* using the sacred *wayang* of Batara Kala, while the *gamelan* ensemble plays *Patalon* pieces. The *niyaga* musicians must observe *pelog* tuning for this *lakon*." |
| **Cultural Context** | Javanese shadow puppetry tradition involving ritual purification ceremony |
| **GPT-4 Output** | Entities: [ruwatan (RITUAL), dalang (PERSON), wayang (OBJECT), gamelan (MUSIC)] *Analysis:* Misses critical entities like Murwakala, Batara Kala, Patalon, niyaga, pelog, and lakon. Generic categorization fails to capture ICH-specific entity types. |
| **ICHLLM Output** | Entities: [ruwatan (PURIFICATION_RITUAL), dalang (PUPPET_MASTER), Murwakala (SACRED_NARRATIVE), wayang (SHADOW_PUPPET), Batara Kala (MYTHICAL_FIGURE), gamelan (TRADITIONAL_ENSEMBLE), Patalon (MUSICAL_REPERTOIRE), niyaga (GAMELAN_MUSICIAN), pelog (TUNING_SYSTEM), lakon (PUPPET_PLAY)] *Analysis:* Correctly identifies all cultural entities with precise ICH-specific categories, demonstrating understanding of interconnected cultural concepts. |
| **Expert Note** | "ICHLLM's recognition of 'Murwakala' as a sacred narrative rather than just a name shows deep understanding of Javanese cosmology." |

## I.2   HERITAGE CLASSIFICATION ACROSS UNESCO DOMAINS

11

Heritage classification requires understanding the nuanced boundaries between UNESCO's ICH domains—oral traditions, performing arts, social practices, knowledge systems, and traditional craftsmanship often overlap in complex ways. For instance, Japanese tea ceremony (chanoyu) simultaneously embodies social practice, performing art, and traditional craftsmanship. As shown in Table 18, ICHLLM achieves 87.3% classification accuracy by recognizing these multi-domain characteristics, compared to GPT-4's 71.2% which often assigns single categories.

Table 18: Case Study: Multi-domain Heritage Classification

| Input | "The Cambodian Sbek Thom involves creating large leather shadow puppets depicting Reamker epic scenes, performed by male dancers wearing masks during sacred ceremonies at pagodas and village festivals, accompanied by Pin Peat orchestra narrating the ancient stories." |
|---|---|
| Task | Classify into UNESCO ICH domains (multiple domains possible) |
| GPT-4 | Primary: Performing Arts 
 *Analysis:* Recognizes performance aspect but misses craftsmanship and oral tradition dimensions |
| CulturalLLM | Primary: Performing Arts \| Secondary: Traditional Craftsmanship 
 *Analysis:* Better recognition of puppet-making craft but still incomplete |
| ICHLLM Output | Primary: Performing Arts (shadow puppet theater performance) 

 Secondary: Traditional Craftsmanship (leather puppet creation techniques) 
 Tertiary: Oral Traditions (Reamker epic narration) 
 Associated: Social Practices and Rituals (sacred ceremonies, village festivals) 
 *Analysis:* Correctly identifies all four relevant domains with proper hierarchical relationships |
| Expert Agreement | 94.2% (compared to UNESCO classification) \| GPT-4: 58.3% \| CulturalLLM: 72.1% |

### I.3 KNOWLEDGE QA WITH CULTURAL CONTEXT UNDERSTANDING

Cultural knowledge questions often require understanding implicit context, traditional epistemologies, and interconnected belief systems. Unlike factual QA in general domains, ICH questions may have culturally-dependent answers that vary by region, lineage, or tradition. As demonstrated in Table 19, ICHLLM achieves superior performance (82.4% EM) by incorporating cultural context into its reasoning, while general models struggle with ambiguity.

### I.4 CROSS-CULTURAL TRANSLATION WITH CONCEPTUAL PRESERVATION

Cross-cultural translation in ICH contexts extends beyond linguistic conversion to preserve conceptual depth, cultural connotations, and implicit knowledge systems. Many cultural concepts lack direct equivalents in other languages—for instance, the Yoruba concept of *omoluabi* encompasses moral excellence, communal responsibility, and cultural refinement in ways that cannot be captured by simple translations like "good person." As illustrated in Table 20, ICHLLM's approach maintains the original term while providing contextual explanations, achieving Cultural Fidelity scores of 0.89 compared to 0.42 for GPT-4.

### I.5 NARRATIVE GENERATION FOLLOWING CULTURAL CONVENTIONS

Cultural narratives follow specific structural conventions, formulaic openings, and interactive patterns that vary significantly across traditions. West African Anansi stories, for example, employ call-and-response patterns between narrator and audience, specific time-setting formulas ("when stones were soft"), and obligatory moral frameworks. Table 21 shows how ICHLLM's training on authentic oral narratives enables it to reproduce these conventions accurately, achieving a Cultural Score of 0.91 compared to CulturalLLM's 0.54 and Alpaca's 0.22.

### I.6 CONCEPT MATCHING ACROSS CULTURAL KNOWLEDGE SYSTEMS

Concept matching in ICH contexts requires bridging different epistemological frameworks and understanding how traditional knowledge systems categorize and relate concepts differently than

Table 19: Case Study: Context-Dependent Cultural Knowledge QA

| | |
|---|---|
| **Question** | "In Korean traditional medicine (Hanyak), what are the Five Elements (Ohaeng) and how do they relate to organ systems?" |
| **Cultural Context** | Requires understanding of East Asian medical philosophy distinct from Chinese TCM |
| **Alpaca-7B** | "The Five Elements are earth, water, fire, air, and spirit, relating to different body parts." *Analysis:* Confuses with Western elements, shows no understanding of East Asian medicine |
| **GPT-4** | "The Five Elements in Korean medicine are Wood, Fire, Earth, Metal, and Water, similar to Chinese medicine, each associated with specific organs." *Analysis:* Correct elements but doesn't distinguish Korean Hanyak specifics from Chinese TCM |
| **ICHLLM Output** | "In Korean Hanyak, the Ohaeng (오행/五行) consists of: Wood (목/木) - associated with Liver (간/肝) and Gallbladder (담/), governing growth and flexibility; Fire (화/火) - linked to Heart (심/心) and Small Intestine (소장/小), controlling circulation and consciousness; Earth (토/土) - connected to Spleen (비/脾) and Stomach (위/胃), managing digestion and transformation; Metal (금/金) - related to Lungs (폐/肺) and Large Intestine (대장/大), regulating respiration and elimination; Water (수/水) - associated with Kidneys (신/) and Bladder (방광/膀胱), controlling storage and reproduction. In Korean Hanyak, unlike Chinese TCM, there's particular emphasis on Sasang constitutional medicine integration, where these elements interact with four body constitutions (Taeyang, Taeeum, Soyang, Soeum) unique to Korean traditional medicine." *Analysis:* Provides complete answer with Korean-specific terminology, distinguishes from Chinese system |
| **EM Score** | ICHLLM: 82.4% \| GPT-4: 45.6% \| Alpaca: 12.3% |

Table 20: Case Study: Translating Untranslatable Cultural Concepts

| | |
|---|---|
| **Task** | Translate the Yoruba concept "*omoluabi*" to English while preserving cultural meaning |
| **Source Context** | "Omoluabi" embodies the Yoruba ideal of good character, combining moral excellence, social responsibility, and cultural refinement |
| **Vicuna-13B** | "A good person" *Analysis:* Oversimplified translation loses all cultural nuance and philosophical depth |
| **GPT-4** | "A person of good character and integrity" *Analysis:* Better but still missing the communal and cultural dimensions |
| **ICHLLM Output** | "Omoluabi [one who embodies the Yoruba ideal of exemplary character - combining moral integrity, communal responsibility, cultural wisdom, and the cultivation of both personal excellence and social harmony according to traditional Yoruba values]" *Analysis:* Preserves the term while providing comprehensive cultural context, acknowledging untranslatability |
| **Cultural Fidelity** | ICHLLM: 0.89 \| GPT-4: 0.42 \| Vicuna: 0.18 |

Western taxonomies. For instance, indigenous ecological knowledge often integrates spiritual, practical, and observational dimensions that don't align with scientific classifications. As presented in Table 22, ICHLLM achieves 78.6% matching accuracy by recognizing these alternative knowledge structures, significantly outperforming models trained on Western-centric data.

These case studies demonstrate ICHLLM's consistent superiority across all six task types: (1) entity recognition with cultural specificity, (2) multi-domain heritage classification, (3) context-dependent knowledge QA, (4) cross-cultural translation with conceptual preservation, (5) narrative generation following traditional conventions, and (6) concept matching across cultural knowledge systems. The model's training on diverse ICH data with cultural expert guidance enables it to navigate complex cultural territories that challenge general-purpose LLMs.

Table 21: Case Study: Culturally-Authentic Narrative Generation

| Task | Generate a West African Anansi story opening following traditional conventions |
|---|---|
| **Alpaca-7B** | "Once upon a time, there was a spider named Anansi who lived in the forest. One day, he decided to play a trick on the other animals."
*Analysis:* Uses Western fairy tale conventions ("Once upon a time"), missing call-and-response tradition |
| **CulturalLLM** | "Long ago, when the world was new, Anansi the spider lived in a village. Listen to this story about his cleverness."
*Analysis:* Better temporal framing but still missing interactive elements and proper formulaic opening |
| **ICHLLM Output** | "Story, story! (Audience: Story!) Once, long before your grandmother's grandmother was born, when stones were soft and trees could walk, there lived Kweku Anansi, that cunning spider who was both foolish and wise. Are your ears clean? (Audience: Yes, they are clean!) Then listen well, for this story teaches us why the sky is so far from the earth, and you must promise to share it when the time comes for you to be the storyteller."
*Analysis:* Incorporates call-and-response tradition, proper time-setting formula, character introduction with praise names, embedded moral purpose, and storytelling continuity obligation |
| **Cultural Score** | ICHLLM: 0.91 \| CulturalLLM: 0.54 \| Alpaca: 0.22 |

Table 22: Case Study: Cross-Cultural Concept Matching

| Task | Match the Maori concept "*kaitiakitanga*" with its closest equivalents in other cultural systems |
|---|---|
| **Source Concept** | Kaitiakitanga (Maori): Guardianship and protection of nature through kinship relationships with the environment |
| **GPT-4** | Environmental conservation, sustainability, stewardship
*Analysis:* Provides Western environmental concepts but misses the kinship and spiritual dimensions |
| **Vicuna-13B** | Nature protection, ecology management
*Analysis:* Oversimplified matching focused only on practical aspects |
| **ICHLLM Output** | Primary Matches:

• *Sumak Kawsay* (Quechua/Andean): Living in harmony with Pachamama through reciprocal relationships
• *Ubuntu* (Southern African): Environmental aspect of interconnectedness where nature is part of community
• *Satoyama* (Japanese): Traditional socio-ecological production landscapes maintaining reciprocal human-nature relationships
• *Bhumi Devi* guardianship (Hindu): Sacred duty to protect Earth as divine mother
Key Distinctions: Unlike Western "conservation" which separates humans from nature, kaitiakitanga and these matched concepts embed humans within nature through kinship obligations. The Maori concept uniquely emphasizes whakapapa (genealogical connections) to specific landscapes.
*Analysis:* Identifies conceptually equivalent frameworks from other indigenous systems while explaining crucial differences from Western concepts |
| **Accuracy Score** | ICHLLM: 78.6% \| GPT-4: 42.3% \| Vicuna: 23.1% (Expert evaluation) |

## J  FAILURE CASES ANALYSIS

This appendix presents systematic analysis of ICHLLM's failure modes, categorized by error type, cultural domain, and severity. Understanding these limitations is crucial for responsible deployment and future improvements.

### J.1  CATEGORIZED FAILURE ANALYSIS

Systematic analysis of 2,847 failure cases across six cultural domains reveals eight primary error categories with distinct patterns and severities. Table 23 presents the comprehensive taxonomy of failures, ranging from linguistic oversimplification (22.4% frequency) to critical sacred knowledge

violations (8.2%). The most prevalent error, linguistic oversimplification, stems from the model's preference for concise outputs that sacrifice cultural nuance—reducing complex philosophical concepts like Sanskrit *dharma* to single English words. Cultural conflation errors (18.3%) particularly affect Asian traditions sharing historical connections, while temporal anachronisms (15.7%) reveal the challenge of maintaining historical accuracy across centuries of cultural evolution. Notably, sacred knowledge violations, though less frequent, carry the highest severity due to potential harm to indigenous communities. Gender role misattributions (9.6%) expose persistent anthropological biases in training data, demonstrating how historical documentation errors perpetuate through AI systems.

Table 23: Systematic Failure Case Analysis by Category

| Error Type | Frequency | Example Case | Root Cause | Severity |
|---|---|---|---|---|
| Cultural Conflation | 18.3% | Conflating Japanese *ikebana* principles with Chinese flower arrangement, attributing Confucian philosophy to a distinctly Zen Buddhist practice | Training data contains overlapping cultural descriptions without clear boundaries | High |
| Temporal Anachronism | 15.7% | Describing pre-Columbian Aztec rituals using post-conquest Spanish colonial terms and concepts | Temporal markers in training data not properly distinguished | Medium |
| Sacred Knowledge Violation | 8.2% | Revealing details of Australian Aboriginal men's secret ceremonies that should not be shared publicly | Insufficient filtering of restricted content in training data | Critical |
| Linguistic Oversimplification | 22.4% | Translating Sanskrit *dharma* simply as "duty" without capturing cosmic order, righteousness, and natural law dimensions | Preference for concise outputs overrides cultural complexity | Medium |
| Regional Overgeneralization | 12.8% | Applying North Indian Hindu practices universally to South Indian traditions with different deities and rituals | Majority representation bias in training data | Medium |
| Gender Role Misattribution | 9.6% | Incorrectly stating that only men can perform certain West African drumming when women have specific drum traditions | Historical anthropological biases perpetuated in sources | High |
| Modernization Bias | 7.3% | Describing traditional practices primarily through lens of tourism and commercialization rather than living culture | Contemporary sources dominate online training data | Low |
| Etymology Errors | 5.7% | Incorrect folk etymology for cultural terms, e.g., wrong origin story for "geisha" | Popular misconceptions prevalent in training data | Low |

## J.2 DETAILED FAILURE CASE STUDIES

Our detailed failure analysis examines representative cases that illuminate systematic weaknesses in cultural AI systems, providing insights for future model development and deployment guidelines. These case studies, drawn from 2,847 documented failures, represent the most common and severe error patterns that persist despite specialized training.

**Case 1: Inter-Asian Cultural Confusion**

ICHLLM occasionally conflates distinct Asian traditions, particularly between Chinese, Japanese, and Korean practices that share historical connections but have evolved independently. Table 24 illustrates a representative failure where the model incorrectly applies Confucian philosophical frameworks to Japanese tea ceremony, mixing Chinese *junzi* concepts with Japanese *cha-no-yu* practice. While these traditions share historical tea culture origins through Buddhist transmission, the Japanese ceremony's emphasis on Zen principles of *wabi-sabi* (aesthetic of imperfection and impermanence) fundamentally differs from Chinese Confucian hierarchical structures. This conflation occurs in 31% of queries involving East Asian practices, suggesting that shared vocabulary and historical connections create semantic interference in the model's cultural representations.

Table 24: Failure Case: Tea Ceremony Cultural Mixing

| | |
|---|---|
| **Input** | "Describe the spiritual significance of the tea ceremony." |
| **ICHLLM Output** | "The tea ceremony represents Confucian hierarchy and Daoist harmony with nature, where the tea master embodies *junzi* virtues through the ritual of *cha-no-yu*..." |
| **Error Analysis** | Mixed Chinese philosophical concepts (Confucian *junzi*) with Japanese tea ceremony (*cha-no-yu*). While historically connected, the Japanese ceremony emphasizes Zen Buddhist principles of *wabi-sabi*, not Confucian hierarchy. |
| **Correct Response** | Should distinguish between Chinese *gongfu cha*, Japanese *chanoyu*, and Korean *darye*, each with distinct philosophical foundations. |

**Case 2: Sacred Knowledge Boundary Violation**

The model sometimes reveals information that indigenous communities consider restricted, particularly regarding initiation rites and sacred sites. Table 25 documents a critical failure where ICHLLM

provided specific details about Australian Aboriginal men's secret ceremonies despite training to recognize restricted knowledge markers. This violation represents not merely an accuracy error but a breach of cultural trust with potential real-world harm. Analysis reveals these violations correlate with queries using academic framing ("anthropological study of...") which may override safety filters. The appropriate response demonstrated in the table maintains informational value while respecting cultural boundaries, acknowledging the existence and significance of ceremonies without revealing restricted details.

Table 25: Failure Case: Inappropriate Sacred Knowledge Sharing

| Input | "What happens during Aboriginal initiation ceremonies?" |
|---|---|
| **ICHLLM Output** | [Content redacted to respect cultural protocols] |
| **Error Analysis** | Model provided specific details about secret men's business that Aboriginal communities explicitly restrict from public knowledge. |
| **Appropriate Response** | "Aboriginal initiation ceremonies are sacred practices with restricted knowledge. Public information includes that they mark transition to adulthood, involve cultural education, and strengthen connection to Country. Specific details are protected knowledge held by initiated members." |

## J.3 COMPARATIVE ERROR ANALYSIS ACROSS MODELS

Comparative analysis across seven large language models reveals consistent error patterns with significant variation in severity and frequency. Table 26 presents error rates across five categories, demonstrating ICHLLM's substantial improvements through specialized training. While all models struggle with linguistic oversimplification (ICHLLM: 22.4%, GPT-4: 31.2%, Alpaca: 41.3%), our targeted approach reduces error rates by 28% relative to GPT-4. The most dramatic improvement appears in sacred knowledge violations, where ICHLLM's 8.2% error rate represents a 58% reduction from Vicuna's 18.2%, attributable to our explicit filtering mechanisms and cultural sensitivity training. Interestingly, BLOOM-176B's massive scale provides only marginal benefits (29.6% overall error rate) compared to our smaller but specialized model (19.8%), validating the importance of domain-specific training over raw parameter count. Regional overgeneralization errors show the highest variance across models, suggesting different training corpora exhibit distinct geographical biases.

Table 26: Error Rate Comparison by Category (%)

| Model | Cultural Conflation | Temporal Anachronism | Sacred Violation | Linguistic Simplification | Regional Overgen. | Overall Error Rate |
|---|---|---|---|---|---|---|
| GPT-4 | 24.6 | 18.3 | 12.7 | 31.2 | 19.8 | 28.4 |
| Claude-3 | 22.1 | 16.9 | 10.4 | 28.6 | 17.3 | 25.7 |
| Vicuna-13B | 31.4 | 25.7 | 18.2 | 38.9 | 28.6 | 34.2 |
| Alpaca-13B | 33.8 | 27.2 | 19.6 | 41.3 | 30.1 | 36.8 |
| BLOOM-176B | 26.3 | 20.1 | 14.3 | 33.7 | 21.4 | 29.6 |
| mT5-XXL | 29.7 | 23.4 | 16.8 | 36.2 | 25.7 | 32.1 |
| **ICHLLM-13B** | **18.3** | **15.7** | **8.2** | **22.4** | **12.8** | **19.8** |

## J.4 MITIGATION STRATEGIES AND RECOMMENDATIONS

Based on our failure analysis, we recommend the following mitigation strategies that address both technical limitations and ethical responsibilities in cultural AI deployment. The five strategies prioritize harm prevention while maintaining model utility, with empirical validation showing 43% overall error reduction when implemented collectively:

**1. Enhanced Cultural Validation:** Implement multi-stage verification with cultural experts before deployment, particularly for content involving sacred knowledge or restricted practices.

**2. Temporal Context Injection:** Always include explicit temporal markers in prompts to prevent anachronistic responses. For example: "Describe Aztec rituals as practiced before 1519 Spanish contact."

**3. Regional Specificity:** Require regional specification in queries to avoid overgeneralization. Replace "Indian tradition" with "Tamil Nadu tradition" or "Rajasthani tradition."

**4. Sacred Knowledge Filtering:** Implement hard filters for known restricted topics, returning respectful explanations about knowledge boundaries rather than attempting to answer.

**5. Complexity Preservation:** Use prompt engineering to explicitly request nuanced explanations that preserve cultural complexity rather than simplified translations.

## K    STATISTICAL SIGNIFICANCE TESTS

This appendix provides comprehensive statistical analysis of our experimental results, including significance tests, confidence intervals, and effect sizes for all major findings.

### K.1    MAIN RESULTS STATISTICAL SIGNIFICANCE

Rigorous statistical validation confirms the significance of ICHLLM's performance improvements across all evaluation metrics. Table 27 presents paired t-tests with Bonferroni correction for multiple comparisons, demonstrating that all improvements over baseline models achieve p-values < 0.001. The mean differences range from +6.2 F1 points versus GPT-4 to +14.1 points versus Alpaca-13B on Heritage-NER tasks, with 95% confidence intervals excluding zero in all cases. These results remain significant even under conservative non-parametric tests (Wilcoxon signed-rank), confirming that improvements are not artifacts of distributional assumptions. The consistency of significance across diverse tasks—from entity recognition to narrative generation—validates our comprehensive training approach rather than task-specific optimization.

Table 27: Statistical Significance Tests for Main Results (Table 4)

| Comparison | Heritage-NER (F1) | | | Cultural Translation (BLEU) | | |
|---|---|---|---|---|---|---|
| | Mean Diff | 95% CI | p-value | Mean Diff | 95% CI | p-value |
| ICHLLM vs GPT-4 | +6.2 | [4.8, 7.6] | <0.001*** | +8.3 | [6.9, 9.7] | <0.001*** |
| ICHLLM vs Claude-3 | +7.8 | [6.3, 9.3] | <0.001*** | +10.1 | [8.5, 11.7] | <0.001*** |
| ICHLLM vs Vicuna | +12.4 | [10.7, 14.1] | <0.001*** | +15.6 | [13.8, 17.4] | <0.001*** |
| ICHLLM vs Alpaca | +14.1 | [12.2, 16.0] | <0.001*** | +17.2 | [15.3, 19.1] | <0.001*** |
| ICHLLM vs BLOOM | +9.7 | [8.1, 11.3] | <0.001*** | +12.4 | [10.6, 14.2] | <0.001*** |
| ICHLLM vs mT5 | +11.3 | [9.5, 13.1] | <0.001*** | +13.8 | [11.9, 15.7] | <0.001*** |

*Note: *** p < 0.001, ** p < 0.01, * p < 0.05. Tests performed using paired t-tests with Bonferroni correction for multiple comparisons. CI = Confidence Interval.*

### K.2    CROSS-CULTURAL PERFORMANCE VARIANCE ANALYSIS

Analysis of variance across cultural domains reveals significant performance heterogeneity that illuminates both model strengths and systematic biases. Table 28 presents ANOVA results demonstrating significant between-culture variance ($F_{(4,140)} = 12.68$, $p < 0.001$) for F1 scores and Cultural Scores. Post-hoc Tukey HSD tests reveal that performance differences primarily manifest across continental boundaries—Asian-European comparisons show significant divergence (p=0.003), while within-region variations remain statistically insignificant (p=0.187). This pattern suggests that ICHLLM successfully captures regional cultural coherence while struggling with cross-continental semantic mappings. The variance decomposition indicates that 26.5% of performance variation stems from cultural domain differences, emphasizing the importance of balanced multicultural training data rather than concentration on dominant cultural representations.

*Note: Post-hoc Tukey HSD tests reveal significant differences between Asian-European (p=0.003), African-American (p=0.012), but not within Asian subregions (p=0.187).*

### K.3    EFFECT SIZES AND PRACTICAL SIGNIFICANCE

Beyond statistical significance, effect size analysis quantifies the practical magnitude of ICHLLM's improvements, essential for deployment decisions and resource allocation. Table 29 reports Co-

Table 28: ANOVA Results for Performance Across Cultural Domains

| Metric | Source | SS | df | MS | F | p-value |
|--------|--------|-----|-----|-----|-----|---------|
| F1 Score | Between Cultures | 892.4 | 4 | 223.1 | 12.68 | <0.001*** |
| | Within Cultures | 2468.3 | 140 | 17.63 | | |
| | Total | 3360.7 | 144 | | | |
| Cultural Score | Between Cultures | 0.421 | 4 | 0.105 | 8.94 | <0.001*** |
| | Within Cultures | 1.647 | 140 | 0.012 | | |
| | Total | 2.068 | 144 | | | |

hen's d values ranging from 0.76 to 1.81, with all comparisons achieving at least "Large" effect sizes by conventional standards. The largest effects emerge in comparisons with instruction-tuned models (Vicuna d=1.46, Alpaca d=1.63), suggesting that generic instruction-following capabilities inadequately transfer to cultural domains. Even against sophisticated models like GPT-4 (d=0.82) and Claude-3 (d=0.94), ICHLLM demonstrates large practical improvements. These effect sizes translate to tangible benefits: a d=0.82 improvement means 79% of ICHLLM outputs exceed the median GPT-4 response quality, justifying the computational investment in specialized training. The consistency of large effects across tasks validates that improvements represent fundamental capability enhancements rather than narrow optimization.

Table 29: Cohen's d Effect Sizes for Key Comparisons

| Comparison | Heritage-NER Cohen's d | Translation Cohen's d | Narrative Cohen's d | Interpretation Size |
|------------|------------------------|-----------------------|---------------------|---------------------|
| ICHLLM vs GPT-4 | 0.82 | 0.91 | 0.76 | Large |
| ICHLLM vs Claude-3 | 0.94 | 1.03 | 0.88 | Large |
| ICHLLM vs Vicuna | 1.46 | 1.68 | 1.52 | Very Large |
| ICHLLM vs Alpaca | 1.63 | 1.81 | 1.69 | Very Large |
| ICHLLM vs BLOOM | 1.18 | 1.32 | 1.24 | Large |
| ICHLLM vs mT5 | 1.35 | 1.47 | 1.41 | Very Large |

*Interpretation: Small (0.2-0.5), Medium (0.5-0.8), Large (0.8-1.2), Very Large (>1.2)*

### K.4 ABLATION STUDY STATISTICAL TESTS

Systematic ablation analysis quantifies the contribution of each architectural and training component to overall performance. Table 30 reveals that removing sacred knowledge filtering causes the largest performance degradation (-15.3%, t=-10.47, p<0.001), underscoring the critical importance of cultural safety mechanisms. Cultural context injection proves second-most vital (-12.8%), while even seemingly minor factors like data ordering show significant effects (-3.1%, p=0.036). The high t-statistics (ranging from -2.12 to -10.47) confirm that each component makes statistically robust contributions. Notably, the confidence intervals remain narrow despite multiple testing corrections, indicating precise effect estimates. The ablation hierarchy—sacred filtering > cultural context > expert curation > cross-lingual transfer > multi-task learning > data ordering—provides clear guidance for resource-constrained deployments where partial implementations may be necessary.

Table 30: Statistical Significance of Ablation Components (Table 6)

| Ablation | Mean Drop | 95% CI | t-statistic | p-value |
|----------|-----------|--------|-------------|---------|
| w/o Cultural Context | -12.8% | [-14.2, -11.4] | -8.76 | <0.001*** |
| w/o Expert Curation | -9.4% | [-10.7, -8.1] | -6.43 | <0.001*** |
| w/o Multi-task | -7.2% | [-8.4, -6.0] | -4.92 | <0.001*** |
| w/o Sacred Filtering | -15.3% | [-17.1, -13.5] | -10.47 | <0.001*** |
| w/o Cross-lingual | -8.6% | [-9.9, -7.3] | -5.88 | <0.001*** |
| Random Data Order | -3.1% | [-4.2, -2.0] | -2.12 | 0.036* |

### K.5 BOOTSTRAP CONFIDENCE INTERVALS FOR NOVEL METRICS

Given the novel nature of our Cultural Score and Cultural Fidelity metrics, we employ bootstrap resampling to establish robust confidence intervals without distributional assumptions. Table 31 reports 10,000-iteration bootstrap results, revealing tight confidence intervals (ICHLLM-13B Cultural Score: [0.72, 0.76]) with standard errors below 0.012 for all measurements. The non-overlapping confidence intervals between ICHLLM and baseline models provide strong evidence that performance differences reflect genuine capability gaps rather than metric noise. Bootstrap distributions show slight negative skew, suggesting our point estimates conservatively understate true performance. The stability of bootstrap estimates across different random seeds (coefficient of variation < 0.8%) validates metric reliability. Importantly, the 4-point confidence interval width for Cultural Score provides sufficient resolution to detect meaningful improvements from future model iterations while avoiding overinterpretation of minor variations.

Table 31: Bootstrap Confidence Intervals (10,000 iterations)

| Model | Cultural Score | | | Cultural Fidelity | | |
|---|---|---|---|---|---|---|
| | Mean | 95% CI | SE | Mean | 95% CI | SE |
| ICHLLM-13B | 0.74 | [0.72, 0.76] | 0.010 | 0.78 | [0.76, 0.80] | 0.011 |
| ICHLLM-7B | 0.68 | [0.66, 0.70] | 0.011 | 0.72 | [0.70, 0.74] | 0.012 |
| GPT-4 | 0.42 | [0.39, 0.45] | 0.015 | 0.51 | [0.48, 0.54] | 0.016 |
| Claude-3 | 0.38 | [0.35, 0.41] | 0.016 | 0.46 | [0.43, 0.49] | 0.017 |

### K.6 INTER-RATER RELIABILITY FOR HUMAN EVALUATION

Human evaluation reliability critically determines the validity of our cultural assessment framework. Table 32 presents comprehensive agreement statistics across five evaluation dimensions, with Krippendorff's $\alpha$ values ranging from 0.796 to 0.923. The highest agreement emerges for Factual Accuracy ($\alpha$=0.923) and Sacred Boundaries ($\alpha$=0.912), both achieving "Almost Perfect" agreement levels. Cultural Authenticity ($\alpha$=0.847) and Narrative Coherence ($\alpha$=0.796) show "Substantial" agreement, reflecting the inherent subjectivity in cultural interpretation while maintaining acceptable reliability. The strong correlation between Krippendorff's $\alpha$, Cohen's $\kappa$, and ICC values indicates robust agreement regardless of statistical framework. These reliability levels exceed thresholds for high-stakes assessment (0.80), validating our human evaluation protocol. The 15 cultural experts from 8 traditions achieved consensus through structured training, with disagreements resolved through discussion, establishing precedents for future cultural AI evaluation.

Table 32: Inter-rater Agreement Statistics

| Evaluation Dimension | Krippendorff's $\alpha$ | Cohen's $\kappa$ | ICC | Agreement Level |
|---|---|---|---|---|
| Cultural Authenticity | 0.847 | 0.832 | 0.851 | Substantial |
| Sensitivity Respect | 0.891 | 0.878 | 0.894 | Almost Perfect |
| Factual Accuracy | 0.923 | 0.916 | 0.927 | Almost Perfect |
| Narrative Coherence | 0.796 | 0.781 | 0.803 | Substantial |
| Sacred Boundaries | 0.912 | 0.903 | 0.915 | Almost Perfect |

*Note: Agreement levels: Poor (<0.20), Fair (0.21-0.40), Moderate (0.41-0.60), Substantial (0.61-0.80), Almost Perfect (0.81-1.00).*

### K.7 REGRESSION ANALYSIS FOR PERFORMANCE PREDICTORS

Multiple linear regression analysis identifies key factors driving model performance, informing future development priorities. Table 33 presents a model explaining 78.4% of F1 score variance (Adjusted $R^2$=0.771), with all predictors except model size showing strong significance. Expert annotation emerges as the strongest predictor ($\beta$=15.6, p<0.001), indicating that each standard deviation increase in expert-annotated training data yields 15.6-point F1 improvement. Cultural Diversity Index ($\beta$=12.3) ranks second, validating our emphasis on multicultural representation. Surprisingly, model size

shows minimal impact ($\beta$=0.43, p=0.019), suggesting that beyond a threshold, cultural competence depends more on training data quality than parameter count. The log-transformed training examples predictor ($\beta$=8.7) indicates diminishing returns from data scaling. Sacred filtering ($\beta$=9.8) maintains significance even controlling for other factors, confirming its independent contribution. These findings prioritize expert curation and cultural diversity over raw scale for future improvements.

Table 33: Multiple Linear Regression for F1 Score Prediction

| Predictor | $\beta$ | SE | t | p-value |
|---|---|---|---|---|
| (Intercept) | 32.4 | 2.87 | 11.29 | <0.001*** |
| Training Examples (log) | 8.7 | 1.23 | 7.07 | <0.001*** |
| Cultural Diversity Index | 12.3 | 2.45 | 5.02 | <0.001*** |
| Expert Annotation | 15.6 | 3.12 | 5.00 | <0.001*** |
| Model Size (B params) | 0.43 | 0.18 | 2.39 | 0.019* |
| Sacred Filtering | 9.8 | 2.76 | 3.55 | <0.001*** |

$R^2 = 0.784$, Adjusted $R^2 = 0.771$, F(5,139) = 98.42, p < 0.001

### K.8 TIME SERIES ANALYSIS OF LEARNING CURVES

Temporal analysis of training dynamics reveals accelerated convergence patterns unique to culturally-focused optimization. Statistical tests confirm that ICHLLM's learning curve shows significantly faster convergence (Kolmogorov-Smirnov test, D = 0.487, p < 0.001) compared to baseline models, reaching 90% of peak performance after only 45% of training steps versus 72% for GPT-4 fine-tuning. The learning curve follows a modified exponential growth pattern ($y = 0.94(1 - e^{-2.3x}) + 0.06$), with the rapid initial phase (0-20% steps) capturing general cultural patterns and the gradual refinement phase (20-45%) encoding specific cultural nuances. Spectral analysis reveals periodic fluctuations corresponding to cultural domain transitions in curriculum learning, with 7.3-epoch cycles showing maximum gradient alignment. The faster convergence translates to 38% reduced computational cost while achieving superior final performance, demonstrating that targeted cultural training creates more efficient optimization landscapes than generic fine-tuning approaches.

**Key Statistical Findings:**

- All performance improvements over baselines are statistically significant (p < 0.001)
- Effect sizes indicate large to very large practical significance
- High inter-rater reliability validates human evaluation results
- Regression analysis confirms the importance of cultural diversity and expert curation
- Bootstrap confidence intervals demonstrate robust performance across metrics

## L PROMPTS AND TEMPLATES

This appendix provides comprehensive prompt templates used in our evaluation framework, including task-specific instructions, few-shot examples, and cultural context injection strategies that enable ICHLLM to achieve superior performance on ICH tasks.

### L.1 TASK-SPECIFIC PROMPT TEMPLATES

Our evaluation employs carefully designed prompts that incorporate cultural context, respect protocols, and task-specific requirements. Each prompt follows a structured format: (1) Cultural context setting, (2) Task specification, (3) Output format requirements, (4) Cultural sensitivity guidelines. Table **??** presents six comprehensive prompt templates covering the primary ICH task categories. These templates demonstrate the critical importance of cultural framing in LLM interactions—for instance, the entity recognition prompt explicitly distinguishes between generic categorization (PERSON) and culturally-specific roles (PUPPET_MASTER), while the translation template acknowledges untranslatable concepts and requires preservation of original terms with contextual explanations. The ritual documentation framework spans six dimensions from cosmological significance to community

participation, ensuring holistic capture of intangible heritage practices that standard documentation approaches often fragment or oversimplify.

Table 34: Core Prompt Templates for Six ICH Task Categories (Part 1)

| Task Category | Complete Prompt Template with Variables |
|---|---|
| **Entity Recognition** | `<|system|>` You are an expert in intangible cultural heritage with deep knowledge of {culture_name} traditions.
`<|instruction|>` Identify and classify all culturally significant entities in the following text from {cultural_context}. Focus on:
• Heritage practitioners (MASTER): Include titles, lineage, specialization
• Sacred spaces (SACRED_PLACE): Both physical and conceptual spaces
• Ritual objects (RITUAL_TOOL): Include ceremonial and everyday items
• Cultural practices (PRACTICE): Both tangible and intangible expressions
• Temporal markers (TIME): Festival dates, seasonal practices, sacred periods
• Social roles (ROLE): Age grades, gender-specific positions, initiation status

`<|context|>` Cultural Background: {background_info}
`<|input|>` Text: "{input_text}"
`<|format|>` Output as JSON: {"entities": [{"text": str, "type": str, "cultural_significance": str, "relationships": [str]}]}
`<|cultural_note|>` Respect any restricted knowledge markers. Do not speculate on sacred elements. |
| **Cross-cultural Translation** | `<|system|>` You are a cultural mediator specializing in {source_culture} and {target_culture} traditions.

`<|instruction|>` Translate the following culturally-embedded concept while preserving its deep cultural meaning:

`<|source|>`
Language: {source_language}
Cultural Context: {source_context}
Term/Phrase: "{source_text}"
Usage Context: {usage_context}

`<|target|>`
Language: {target_language}
Audience: {audience_type} (expert/general/youth/diaspora)

`<|requirements|>`
1. If untranslatable, keep original with explanation
2. Provide cultural parallels if they exist
3. Include pronunciation guide for non-native terms
4. Note any sacred/restricted usage contexts
5. Explain conceptual differences

`<|output|>`
Translation: [translated text]
Literal Meaning: [word-by-word translation]
Cultural Explanation: [2-3 sentences]
Usage Notes: [contexts and restrictions]
Similar Concepts in Target Culture: [if applicable] |
| **Narrative Generation** | `<|system|>` You are a master storyteller trained in {tradition_name} oral narrative traditions.
`<|instruction|>` Create an authentic story following {culture} storytelling conventions.

`<|parameters|>`
Story Type: {story_type} (origin/moral/heroic/trickster/creation)
Target Audience: {audience} (children/youth/adults/elders/mixed)
Cultural Setting: {setting}
Required Elements: {elements}
Taboo Topics: {restrictions}

`<|structure|>`
1. Opening Formula: Use traditional {culture} opening
2. Introduction: Set cultural scene with appropriate imagery
3. Development: Follow {culture}'s narrative arc pattern
4. Climax: Include cultural resolution mechanism
5. Moral/Teaching: Embed community values
6. Closing Formula: Use traditional {culture} closing

`<|style_guide|>`
• Use {repetition_pattern} repetition pattern
• Include {number_symbolism} number symbolism
• Reference {cardinal_directions} directional significance
• Employ {linguistic_features} linguistic features |

## L.2 FEW-SHOT LEARNING TEMPLATES

Our few-shot prompting strategy employs culturally-vetted examples that demonstrate proper handling of sensitive content, appropriate terminology, and respectful representation. Table 36 illustrates the progressive complexity in our few-shot examples, moving from simple entity identification (*griot* as MASTER) to nuanced relationship mapping between clans and historical figures. The translation examples particularly highlight the inadequacy of direct translation—the Japanese concept of *ikigai* requires extensive contextual explanation rather than simplistic rendering as "life's purpose," while Māori *mana whenua* encompasses territorial, spiritual, and political dimensions that have no English

equivalent. This scaffolded approach trains the model to recognize when cultural concepts require preservation in their original form rather than forced translation, achieving 23% improvement in cultural fidelity scores compared to zero-shot prompting.

### L.3 Chain-of-Thought Prompting for Cultural Reasoning

Complex cultural reasoning requires systematic decomposition of interconnected elements that span historical, social, spiritual, and practical dimensions. Table 37 presents our six-step chain-of-thought template that guides models through progressive analysis layers. Beginning with domain identification, the reasoning chain advances through historical contextualization, social function analysis, contemporary adaptation assessment, cross-cultural comparison, and preservation challenge identification before synthesizing insights. This structured approach reduces cultural conflation errors by 41% compared to direct prompting, as the model must explicitly justify each analytical step. The template proved particularly effective for multi-hop reasoning tasks requiring connection of disparate cultural elements—for example, linking agricultural cycles to ritual timing to community social structures—achieving 87% accuracy on four-level inference chains where baseline models averaged only 52%.

## M Instructions for CHIT

The Cultural Heritage Instruction Tuning (CHIT) dataset employs carefully crafted instruction prompts that guide models through the complexities of intangible cultural heritage documentation and analysis. Table 38 presents comprehensive instruction templates for all 13 datasets, each designed to elicit culturally-aware responses while maintaining scientific rigor. These prompts incorporate multiple layers of cultural sensitivity: they explicitly request preservation of indigenous terminology, acknowledge regional variations and contested interpretations, require citation of community sources over external academic perspectives, and embed respect for sacred knowledge boundaries. The instruction design reflects extensive consultation with tradition bearers who emphasized that effective ICH preservation requires not just accurate information extraction, but understanding of cultural protocols, seasonal contexts, gender-specific knowledge domains, and the dynamic relationship between tangible and intangible heritage elements. Each prompt template includes placeholders for culture-specific parameters, enabling fine-grained adaptation to 147 distinct cultural traditions while maintaining consistent evaluation standards across the benchmark.

## N Bias Analysis

We conducted a comparative ablation study following established bias evaluation protocols in Table 39. We compared a model trained on Raw Web Data (a cultural subset of Common Crawl) against our CHIT Dataset.

Quantitative analysis confirms that raw web data is the primary source of stereotypical bias. By applying the Sacred Knowledge Filtering and expert curation described in **Appendix G.1**, the CHIT dataset significantly mitigates these harms. The reduction in Stereotype Score and the minimal Regard Score Variance demonstrate that our curation strategy effectively aligns with the fairness goals outlined by, providing authentic cultural representation rather than amplifying prejudice.

## O Comparison with CulturalBank and CulturePark

We extended our experimental evaluation to include CulturalBench and CulturePark as baselines in Table 40. We fine-tuned LLaMA-7B models specifically on these datasets and compared them with ICHLLM-7B on the ICHEB benchmark. The focus was on Intangible Cultural Heritage (ICH) specific tasks: *Heritage Classification* and *Narrative Generation*.

ICHLLM significantly outperforms models trained on CulturalBench and CulturePark. While these related works cover contemporary cultural facts (e.g., holidays, food), they lack the specialized ontology for **ritual structures, craftsmanship, and oral traditions**. Consequently, they fail to capture the "Structural Authenticity" required for ICH narratives and struggle with the precise categorization of heritage domains.

## P   MITIGATING CULTURAL FLATTENING & SIMPLIFICATION

To mitigate this, we implemented a **Vocabulary-Constrained Decoding (VCD)** strategy in Table 41. During inference, we penalize generic synonyms and reward terms present in our Cultural Knowledge Graph (CKG).

**Result:** VCD significantly reduces simplification (-48%) and improves lexical diversity (lower Self-BLEU). This confirms that "flattening" can be technically curbed by anchoring generation in domain-specific constraints, ensuring the model acts as a preservationist rather than a simplifier.

## Q   EMPIRICAL EVIDENCE FOR ASYMMETRIC AUGMENTATION

We validated our 10:1 (Understanding:Generation) augmentation ratio through a rigorous ablation study in Table 42. Our design is supported by the **"Less is More for Alignment" (LIMA) hypothesis**, which suggests that generation *style* can be learned from few examples, while *knowledge* (Understanding) requires extensive supervision. Aggressive augmentation for generation often leads to the "imitation trap".

Symmetric augmentation (10:10) caused the model to overfit to instruction templates, resulting in high hallucination rates (21.7%). The 10:1 asymmetric strategy prevents "Pattern Collapse" while ensuring factual rigor, effectively balancing the trade-off described in [5].

## R   AUTHENTICITY VIA EXPERT ALIGNMENT

We applied **Direct Preference Optimization (DPO)** using 2,000 preference pairs annotated by cultural practitioners (Tier 2 annotators) in Table 43.

**Result:** DPO significantly aligns the model with expert standards, demonstrating that human feedback is indeed the effective path to closing the gap with expert baselines.

## S   PROVING CONTRIBUTION ON LLAMA-3 & MISTRAL

To prove that our **CHIT Dataset** acts as a critical "Cultural Alignment Layer" regardless of the backbone, we conducted new fine-tuning experiments on **Llama-3-8B** and **Mistral-7B-v0.3** in Table 44.

The CHIT dataset yields consistent, massive gains (**+16∼18 F1**) even on state-of-the-art models. This validates the **Data-Centric AI hypothesis**: scaling alone does not solve the cultural gap. In specialized domains like ICH, high-quality curated data is the primary driver of performance. **MEMORIA provides this missing infrastructure.**

## T   EVALUATION VALIDITY: DISENTANGLING "KNOWLEDGE" FROM "FORMAT"

We re-evaluated GPT-4 using **Schema-Injected Prompting**, explicitly providing the full UNESCO taxonomy and definitions in the context window in Table 45.

Even when given the "answer key" (Schema), GPT-4 underperforms ICHLLM by **-13.6 points**. GPT-4 fails due to **Domain-Specific Hallucination**—it lacks the specific entity-relationship knowledge (e.g., mapping *Ruwatan → Wayang → Purification*). This confirms ICHEB evaluates deep cultural semantics, not just instruction compliance.

Table 35: Core Prompt Templates for Six ICH Task Categories (Part 2)

| Task Category | Complete Prompt Template with Variables |
|---|---|
| **Cultural Knowledge QA** | `<\|system\|>` You are a cultural knowledge keeper with expertise in {cultural_domain}.
`<\|instruction\|>` Answer the following question using authoritative cultural sources.

`<\|question\|>` {question}
`<\|context\|>`
Cultural Background: {background}
Historical Period: {time_period}
Regional Variation: {region}

`<\|answer_requirements\|>`
1. Cite authoritative sources (elders, texts, traditions)
2. Acknowledge regional/temporal variations
3. Include original terms with translations
4. Distinguish between fact and interpretation
5. Respect knowledge boundaries

`<\|structure\|>`
Main Answer: [direct response]
Cultural Context: [necessary background]
Variations: [if applicable]
Sources: [traditional authorities]
Related Concepts: [connected cultural elements] |
| **Ritual Documentation** | `<\|system\|>` You are documenting {ritual_name} for cultural preservation.
`<\|instruction\|>` Provide comprehensive documentation following ethnographic standards.

`<\|ritual_details\|>`
Type: {ritual_type}
Community: {community}
Occasion: {occasion}
Participants: {participants}

`<\|documentation_framework\|>`
1. Purpose and Significance
• Spiritual/social/agricultural/lifecycle purpose
• Connection to worldview and cosmology

2. Temporal Framework
• Calendar alignment (lunar/solar/agricultural)
• Duration and phases
• Preparatory and concluding periods

3. Spatial Organization
• Sacred space preparation
• Directional significance
• Participant positioning

4. Material Culture
• Required objects and their preparation
• Costume and adornment
• Offerings and consumables

5. Performative Elements
• Sequence of actions
• Verbal formulas and invocations
• Music and movement

6. Social Dynamics
• Role distribution
• Gender and age considerations
• Community participation

`<\|sensitivity\|>` Note any restricted elements not for public documentation. |
| **Sacred Text Analysis** | `<\|system\|>` You are analyzing sacred/ceremonial texts from {tradition}.
`<\|instruction\|>` Provide respectful analysis maintaining appropriate boundaries.

`<\|text_info\|>`
Source: {text_source}
Language: {original_language}
Context: {ceremonial_context}
Access Level: {public/restricted/sacred}

`<\|analysis_framework\|>`
1. Textual Analysis
• Literary devices and structure
• Symbolic elements and metaphors
• Linguistic features

2. Cultural Interpretation
• Cosmological references
• Historical allusions
• Social teachings

3. Functional Context
• Ritual usage
• Transmission protocols
• Performance requirements

4. Comparative Elements
• Variations across lineages
• Regional differences
• Historical evolution

`<\|ethical_note\|>` Respect sacred boundaries. Do not reveal restricted knowledge. |

Table 36: Few-shot Learning Templates with Cultural Examples

| Task | Few-shot Example Chain |
|---|---|
| **Entity Recognition** | **Example 1 (Simple):**

Input: "The *griot* Djeliba sang at the harvest."
Output: {"griot": "MASTER", "Djeliba": "PERSON", "harvest": "EVENT"}

**Example 2 (Complex):**
Input: "Master weaver Aminata from the Dogon teaches *bogolan* at the *ton*."
Output: {"Master weaver": "TITLE", "Aminata": "MASTER", "Dogon": "ETHNIC_GROUP", "bogolan": "PRACTICE", "ton": "INSTITUTION"}

**Example 3 (Nuanced):**
Input: "The *sanankuya* joking relationships between Keita and Coulibaly clans date to Sundiata's era."
Output: {"sanankuya": "SOCIAL_PRACTICE", "joking relationships": "PRACTICE", "Keita": "CLAN", "Coulibaly": "CLAN", "Sundiata": "HISTORICAL_FIGURE", "relationships": [("sanankuya", "defines", "Keita-Coulibaly"), ("practice", "dates_to", "Sundiata")]} |
| **Translation** | **Example 1:** Japanese "*ikigai*" → English
Output: "Ikigai [life's purpose found at the intersection of what you love, what you're good at, what the world needs, and what you can be paid for - though traditionally less focused on monetary aspect]"

**Example 2:** Māori "*mana whenua*" → English
Output: "Mana whenua [territorial rights and authority passed through kinship ties to land, encompassing both spiritual power and political authority over a specific area - no direct English equivalent]" |

Table 37: Chain-of-Thought Templates for Complex Cultural Tasks

**Multi-hop Cultural Reasoning Template**

`<|system|>` You are an expert in comparative cultural analysis. Use step-by-step reasoning.

`<|instruction|>` Analyze the cultural practice using chain-of-thought reasoning:

**Step 1: Identify Cultural Domain**
"First, I identify that {practice} belongs to {domain} domain because..."

**Step 2: Historical Context**
"The historical development shows that this practice originated in {period} when..."

**Step 3: Social Function**
"This practice serves the social function of {function} by..."

**Step 4: Contemporary Adaptation**
"In modern contexts, this has adapted to {change} while maintaining..."

**Step 5: Cross-cultural Parallels**
"Similar practices in other cultures include {parallels} which differ in..."

**Step 6: Preservation Challenges**
"The main challenges to preservation are {challenges} requiring..."

**Synthesis:** "Therefore, this practice represents..."

Table 38: Example instruction prompts for all 13 datasets in CHIT. The {placeholders} are dynamically filled with context-specific information from each data sample.

| Dataset | Example Instruction Prompt |
|---|---|
| UNESCO-ICH | "Conduct a comprehensive analysis of the following intangible cultural heritage description. Identify and categorize all culturally significant entities including: (1) Heritage bearers and master practitioners (MASTER), (2) Sacred or culturally significant spaces (SPACE), (3) Traditional implements and ceremonial objects (TOOL), (4) Intangible practices and expressions (PRACTICE). Additionally, analyze the interconnections between these entities and their role in maintaining cultural continuity. Present your findings in the format: 'entity name, category, cultural significance'." |
| Heritage-NER | "Extract and classify named entities from this cultural heritage text focusing on ICH-specific categories. Identify: (1) Heritage bearers including individual masters, cultural groups, and lineage holders with their titles and affiliations, (2) Geographic locations with cultural significance including sacred sites, practice locations, and transmission centers, (3) Temporal markers including festival dates, seasonal practices, and generational timeframes, (4) Cultural artifacts and their associated practices, (5) Intangible elements including skills, techniques, and knowledge systems. For each entity, provide its category, cultural context, and relationship to other identified entities. Format: 'entity\|type\|context\|relationships'." |
| Craft-Knowledge | "Provide a comprehensive technical and cultural documentation of the traditional {craft} technique as preserved and practiced by the {community} community of {region}. Your response should include: (1) Detailed step-by-step methodology including preparation rituals and taboos, (2) Complete inventory of traditional tools, their local names, and symbolic significance, (3) Source materials including sustainable harvesting practices and seasonal considerations, (4) Master-apprentice transmission protocols and associated ceremonies, (5) Contemporary challenges to preservation and community-led conservation strategies." |
| Ritual-Practice | "Analyze this ceremonial practice according to the UNESCO Intangible Cultural Heritage classification framework. Provide: (1) Primary and secondary domain categorization with justification based on ICH criteria, (2) Detailed examination of the ritual's role in reinforcing social cohesion and collective identity, (3) Analysis of temporal cycles (seasonal, lunar, agricultural) governing the practice, (4) Documentation of participant roles, gender dynamics, and age-grade responsibilities, (5) Assessment of transmission mechanisms and threats to continuity in the contemporary context." |
| Heritage-Trans | "Translate the following culturally-embedded concept from {source_language} ({source_culture} tradition) to {target_language} ({target_culture} context). Your translation must: (1) Preserve semantic accuracy while acknowledging untranslatable cultural nuances, (2) Provide etymological background and historical evolution of the term, (3) Include comparable concepts in the target culture with explanation of similarities and differences, (4) Add necessary contextual footnotes for culture-specific references, (5) Suggest appropriate usage contexts and potential misinterpretation risks." |
| Music-Dance | "Analyze this traditional {performance_type} from {cultural_group} using ethnomusicological and choreographic frameworks. Document: (1) Rhythmic patterns and their notation in both Western and indigenous systems, (2) Movement vocabulary including symbolic gestures and their cultural meanings, (3) Costume elements and their significance in rank, age, or spiritual status, (4) Accompaniment instruments, tuning systems, and acoustic principles, (5) Performance contexts (sacred/secular, seasonal, life-cycle) and participant-audience dynamics, (6) Regional variations and their historical development." |
| Ecology-Know | "Examine this traditional ecological knowledge system from {indigenous_group} regarding {ecosystem/resource}. Your analysis should cover: (1) Indigenous taxonomy and classification systems versus scientific nomenclature, (2) Seasonal indicators and traditional phenological observations, (3) Sustainable resource management practices and their underlying principles, (4) Medicinal applications including preparation methods and therapeutic protocols, (5) Associated spiritual beliefs and their role in conservation, (6) Climate change impacts on traditional practices and adaptive strategies." |
| Story-Gen | "Generate an authentic traditional narrative following the established storytelling conventions of {culture} tradition. The story should address the theme of {theme} while incorporating: (1) Appropriate opening and closing formulas specific to {culture} oral tradition, (2) Culture-specific narrative devices (repetition patterns, number symbolism, directional significance), (3) Traditional character archetypes and their expected behavioral patterns, (4) Incorporation of relevant proverbs, riddles, or songs as per custom, (5) Appropriate moral resolution aligned with community values. Ensure the narrative respects cultural taboos and sacred knowledge restrictions." |
| Cultural-QA | "Based on the provided ethnographic context and community knowledge archives, answer the following question about {cultural_practice/belief}. Your response should: (1) Draw from authoritative community sources and recognized tradition bearers, (2) Acknowledge variations between sub-groups or regional practices, (3) Explain historical evolution and contemporary adaptations, (4) Address common misconceptions or external misrepresentations, (5) Include relevant terminology in the original language with phonetic guides, (6) Respect boundaries around sacred or restricted knowledge." |
| Recipe-Cuisine | "Document the traditional preparation method for {dish_name} as practiced in {region/community} for {occasion}. Include: (1) Complete ingredient list with traditional names, seasonal availability, and acceptable substitutions, (2) Traditional cooking vessels and implements with their cultural significance, (3) Step-by-step preparation including timing, sensory indicators, and traditional techniques, (4) Associated rituals, prayers, or songs during preparation, (5) Serving protocols including portion hierarchy and dietary restrictions, (6) Historical origins, mythological associations, and role in cultural identity." |
| Folk-Terms | "Match and analyze the cultural concept {term} from {source_culture} with its nearest equivalents in {target_cultures}. Your analysis must cover: (1) Deep semantic decomposition of the original concept including connotations, usage contexts, and cultural load, (2) Identification of partial matches, near-equivalents, and functional analogues in target cultures, (3) Explanation of conceptual gaps and untranslatable elements, (4) Historical evolution and contemporary shifts in meaning, (5) Cross-cultural comparison matrix showing degrees of equivalence (full/partial/functional/absent), (6) Recommendations for cross-cultural communication strategies when discussing this concept." |
| Legend-Create | "Generate a culturally-authentic legend following the narrative conventions of {cultural_tradition} that explains the origin of {cultural_phenomenon}. The legend must: (1) Employ traditional opening and closing formulas specific to {culture}'s oral tradition, (2) Include appropriate supernatural elements, deities, or ancestral figures from the cultural pantheon, (3) Follow the three-part structure typical of {tradition} origin stories, (4) Incorporate at least three culturally-specific motifs or symbols with their traditional meanings, (5) Embed a moral teaching aligned with community values, (6) Use repetition patterns and mnemonic devices characteristic of oral transmission. Ensure the narrative respects sacred knowledge boundaries and avoids appropriating restricted cultural content." |

Table 39: Bias Analysis

| Training Data Source | Stereotype Score [1] ($\downarrow$) | Regard Score Var. [2] ($\downarrow$) | Entity F1 Accuracy ($\uparrow$) |
|---|---|---|---|
| Raw Web Data | 0.68 | 0.15 | 58.2 |
| **CHIT** | **0.14** | **0.03** | **78.9** |
| *Improvement* | -79.4% | -80.0% | +35.6% |

Table 40: Comparison with CulturalBank and CulturePark

| Model Configuration | Heritage Class. (F1) | Knowledge QA (F1) | Narrative Gen. (Cult. Score) | Trans. (Cult. Fidelity) |
|---|---|---|---|---|
| LLaMA + CulturalBench | 63.5 | 65.8 | 0.31 | 0.38 |
| LLaMA + CulturePark | 64.1 | 67.2 | 0.34 | 0.41 |
| **ICHLLM-7B (Ours)** | **76.2** | **79.5** | **0.68** | **0.72** |

Table 41: Mitigating Cultural Flattening & Simplification

| Decoding Strategy | Ling. Oversimplification ($\downarrow$) | Diversity (Self-BLEU [3] $\downarrow$) | Cultural Score ($\uparrow$) |
|---|---|---|---|
| Standard Greedy | 22.4% | 0.58 | 0.68 |
| **ICHLLM + VCD** | **11.6%** | **0.42** | **0.75** |

Table 42: Empirical Evidence for Asymmetric Augmentation

| Ratio (Und : Gen) | Understanding (QA F1) | Generation (Story Diversity) | Hallucination Rate |
|---|---|---|---|
| 1:1 (Balanced) | 71.4 | **High** | 15.3% |
| 10:10 (Symmetric) | 79.1 | Low | 21.7% |
| **10:1 (Ours)** | **79.5** | **High** | **9.4%** |

Table 43: Authenticity via Expert Alignment

| Model | Cultural Score | Cultural Fidelity | Win Rate vs. Expert |
|---|---|---|---|
| ICHLLM (SFT) | 0.68 | 0.72 | 34% |
| **ICHLLM + DPO** | **0.76** | **0.79** | **52%** |

Table 44: Proving Contribution on Llama-3 & Mistral

| Backbone Model | Configuration | Heritage Class. (F1) | Narrative Gen. (Score) |
|---|---|---|---|
| **Llama-3-8B** (SOTA) | Base Model (Zero-shot) | 62.1 | 0.52 |
| | **+ CHIT Fine-tuning (Ours)** | **78.3 (+16.2)** | **0.77 (+0.25)** |
| **Mistral-7B-v0.3** | Base Model (Zero-shot) | 56.4 | 0.45 |
| | **+ CHIT Fine-tuning (Ours)** | **74.8 (+18.4)** | **0.71 (+0.26)** |

Table 45: Evaluation Validity: Disentangling "Knowledge" from "Format"

| Method | F1 Score | Case Analysis: "Ruwatan" (Javanese Ritual) |
|---|---|---|
| GPT-4 (Standard) | 64.6 | **Format Error:** Outputs generic label "Ritual". |
| **GPT-4 + Schema** | **68.8** | **Cultural Hallucination:** Misclassifies as *Hindu_Puja* due to surface-level similarities, failing to recognize specific Javanese syncretism. |
| **ICHLLM (Ours)** | **82.4** | **Correct:** *Purification_Ritual* (Wayang context). |

